# EXTENDING TEST-TIME SCALING: A 3D PERSPECTIVE WITH CONTEXT, BATCH, AND TURN

## ABSTRACT

Reasoning reinforcement learning (RL) has recently revealed a new scaling effect: test-time scaling. Thinking models such as R1 and o1 improve their reasoning accuracy at test time as the length of the reasoning *context* increases. However, this effect is fundamentally limited by the context window of the base models, which remains orders of magnitude smaller than the amount of tokens consumed during training. We revisit test-time enhancement techniques through the lens of scaling effect and introduce a unified framework of multi-dimensional test-time scaling to *extend* the capacity of test-time reasoning. Beyond conventional context-length scaling, we consider two additional dimensions: *batch scaling*, where accuracy improves with parallel sampling, and *turn scaling*, where iterative self-refinement enhances reasoning quality. Building on this perspective, we propose 3D test-time scaling, which integrates context, batch, and turn scaling. We show that: (1) each dimension demonstrates a test-time scaling effect, but with a bounded capacity; (2) combining all three dimensions substantially improves the reasoning performance of challenging testbeds, such as IOI, IMO, and CPHO, and further benefits from human preference feedback; and (3) the human-in-the-loop framework naturally extends to a more open-ended domain, i.e., embodied learning, which enables the design of humanoid control behaviors.

## 1 INTRODUCTION

Recent progress in reasoning reinforcement learning has introduced a new form of scaling effect by training thinking models such as R1 (Guo et al., 2025) and o1 (OpenAI, 2024). Unlike conventional models that directly map input to output, a thinking model performs intermediate reasoning computation before producing its final answer. A striking phenomenon emerges during the reinforcement learning process: as the model is trained to reason over progressively longer contexts, its reasoning accuracy steadily improves (Shi et al., 2025; Aggarwal & Welleck, 2025). At test time, this trend continues: extending the reasoning context length consistently leads to higher accuracy. This phenomenon is referred to as test-time scaling of reasoning models (Muennighoff et al., 2025).

However, the potential of test-time scaling is fundamentally constrained by the context window size of current models. Even the most advanced commercial reasoning systems today support fewer than one million tokens of context—negligible compared with the scale of training-time compute, where tens of trillions of tokens are typically consumed during pretraining or post-training. This discrepancy naturally raises a question:

*How should we extend the capacity of test-time scaling?*

Notably, there have been many popular heuristics to enhance the reasoning model's performance at test time. For example, majority voting improves accuracy by generating multiple candidate outputs in parallel and selecting the most frequent one (Wang et al., 2023a). Other approaches, such as Reflexion (Shinn et al., 2023) and in-context learning (Madaan et al., 2023), perform iterative self-refinement, where a model repeatedly revisits and improves its own solutions. Empirically, taking multiple refinement steps leads to a higher accuracy compared with directly outputting the solution.

In this paper, we revisit these diverse techniques within a unified framework of *multi-dimensional test-time scaling*. Specifically, we consider three dimensions: (1) Context scaling: reasoning accuracy improves with longer thinking context lengths; (2) Batch scaling: methods such as majority

vote can be viewed as scaling along a batch dimension, where more parallel samples yield better aggregated answers; (3) Turn scaling: iterative refinement methods correspond to scaling along a turn dimension, where more refinement turns enhance accuracy. Each of these dimensions of scaling interacts with the context-length limits and capabilities of base LLMs, creating unique empirical trade-offs.

Building on this perspective, we propose *3D test-time scaling*, which integrates all three dimensions: context, batch, and turn. We demonstrate that this unified view substantially extends the ceiling of test-time scaling compute and further enables a human-in-the-loop framework that applies to even open-ended domains.

- We establish that each scaling dimension individually exhibits a test-time scaling effect: higher token consumption leads to higher accuracy. However, clear scaling limits can be observed for each dimension.

- We show that the unified 3D test-time scaling is capable of leveraging substantially more tokens for improved reasoning and achieving gold-level performances on challenging Olympiad competition problems, such as IOI, IMO, and CPHO. The framework also extends to a human-in-the-loop setting, where a human operates along the batch dimension and selects the best candidate to further amplify final accuracy.

- Finally, we extend this human-in-the-loop framework to embodied learning, demonstrating that multi-dimensional test-time scaling enables models to interactively design open-ended behaviors in humanoid robot control.

## 2 RELATED WORK

**Scaling Laws.** The cross-entropy loss in large language model pretraining has been shown to scale predictably with key training resources, including model size, dataset size, and compute budget (Kaplan et al., 2020; Rae et al., 2022; Hoffmann et al., 2022). With the emergence of thinking models such as DeepSeek-R1 (Guo et al., 2025) and OpenAI o1 (OpenAI, 2024), researchers investigated scaling laws beyond the number of training tokens. For example, Shi et al. (2025) examines scaling behavior with respect to context length. Scaling laws have also been studied at test-time: Wu et al. (2025); Snell et al. (2024) analyze how performance scales with respect to inference compute under different inference strategies such as majority voting and tree search, as well as tradeoffs between model size and test-time token budgets. In this paper, we focus on test-time scaling and propose a unified framework for characterizing its effects across three dimensions: context scaling, batch scaling, and turn scaling. In contrast, prior work on test-time scaling laws has typically examined only a subset of these aspects.

**Test-Time Scaling.** Test-Time Scaling (TTS) refers to the class of algorithms for improving the model's performance through scaling inference-time compute. TTS methods can be broadly categorized into three approaches. *Context scaling* methods improve performance through longer output sequences, exemplified by Chain-of-Thought prompting (Wei et al., 2023), which elicits step-by-step reasoning in large language models to improve performance on various benchmarks. Recent advances in reasoning models like o1 (OpenAI, 2024) and DeepSeek-R1 (Guo et al., 2025) further incentivize this ability, highlighting context scaling as an effective strategy for improving test-time performance. *Batch scaling* approaches leverage parallel computation to explore multiple reasoning paths. Majority voting is a representative technique that leverages the power of parallel sampling (Wang et al., 2023a) by generating multiple independent reasoning paths and selecting the majority final answer. Other work further incorporates test-time search (Yao et al., 2023), Monte-Carlo tree search (Zhang et al., 2024; Xie et al., 2024), and parallel thinking (Ning et al., 2023) to improve the performance. *Turn scaling* methods improve performance through iterative refinement, including Self-Refine (Madaan et al., 2023), which enables models to iteratively improve outputs through self-feedback without additional training, and Reflexion (Shinn et al., 2023), which reinforces language agents through linguistic feedback and episodic memory to enhance future decision-making.

## 3 FORMULATION OF TEST-TIME SCALING

**LLM Reasoning.** In this work, we focus on LLM reasoning. Given a question $x$, the goal is to derive a correct step-by-step solution $y$. We assume the existence of a ground-truth verifier $\mathcal{R}$ that evaluates the correctness of a solution $y$. This verifier $\mathcal{R}$ could have different implementations depending on the specific case in practice. For example, in mathematical reasoning tasks where the goal is to derive a single numerical answer, this verifier could return a 0-1 score indicating whether the answer in solutions $y$ matches the ground-truth answer. In coding tasks, the score is determined by the set of tests passed by the submitted code in solution $y$. An LLM $\pi_\theta$ is a policy parameterized by $\theta$. Given an input question $x$, the LLM auto-regressively generates an array of tokens one by one. For a dataset of questions $\mathcal{D}$, the expected score of an LLM policy $\pi_\theta$ given a question $x$ is defined as $J(\mathcal{D}, \pi_\theta) = \mathbb{E}_{x \sim \mathcal{D}, y \sim \pi_\theta(\cdot|x)}[\mathcal{R}(x, y)]$.

**Test-Time Scaling.** Test-time scaling approaches aim to achieve a better score through spending more test-time compute. For instance, context scaling allows the LLM to generate longer responses to conduct in-depth exploration. The efficacy of any test-time scaling method must be evaluated along two key aspects: the expected score and the computational cost. In this work, we quantify computational cost using the theoretical maximum number of tokens generated throughout the inference process.

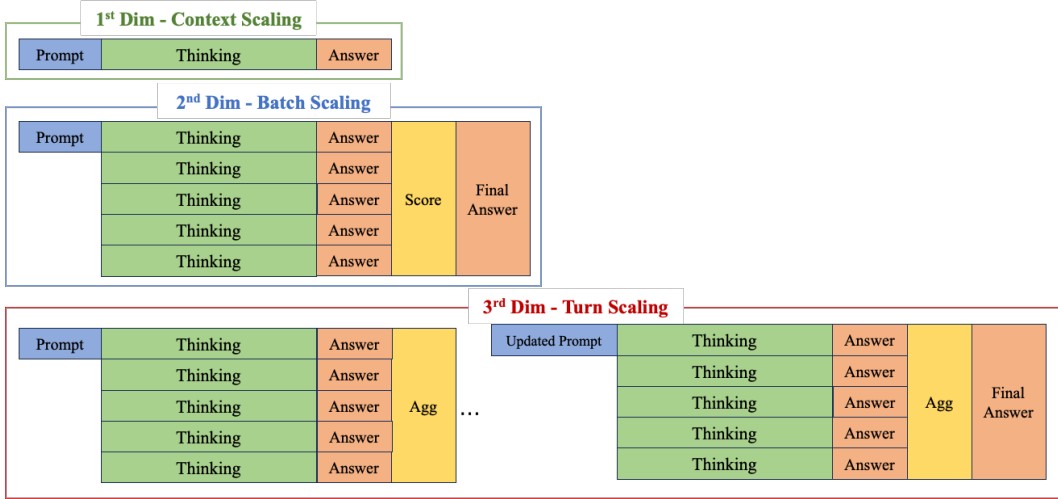

Figure 1: Illustration of Test-time Scaling across three dimensions: context, batch, and turn.

### 3.1 TEST-TIME SCALING WITH CONTEXT, BATCH, AND TURN

**Context Scaling.** We first consider the context dimension. In context scaling, we follow the most straightforward approach of controlling the reasoning process with a maximum token budget $C$. Specifically, when the LLM generates a response $y \sim \pi_\theta(\cdot|x)$ exceeding $C$, we directly truncate the response. Therefore, the expected score of context scaling under a context length $C$ is $J_{\text{context}}(\mathcal{D}, \pi_\theta, C) = \mathbb{E}_{x \sim \mathcal{D}, y \sim \pi_\theta(\cdot|x)}[\mathcal{R}(x, y_{:C})]$

**Batch Scaling.** To scale test-time compute along the batch dimension, notable examples of batch scaling approaches include Majority Voting and Best-of-N (Wang et al., 2023b; Snell et al., 2024; Cobbe et al., 2021). Given the size of the batch dimension $B$, i.e., the number of responses generated in parallel, and the context length $C$, $B$ responses $y_1, y_2, \cdots, y_B$ are first generated under a token budget of $C$. Then, a final solution is derived by selecting the best response through a scoring function $\text{Score}(y)$, i.e. $y_{\text{final}} = \arg\max_i \text{Score}(y_i)$. In cases where the goal is to derive a single numerical answer, the scoring function could be implemented by counting the occurrence of each candidate answer in the $B$ responses. In coding tasks or mathematical proof tasks where it is infeasible to apply voting, we employ the LLM $\pi_\theta$ to select a single solution out of the $B$ responses. In this paper, we refer to majority voting as **Batch Scaling (Vote)** and the Best-of-$N$ strategy as **Batch Scaling (Best-of-$N$).**

**3D Scaling with a Turn Dimension.** We introduce an additional turn axis to batch scaling, leading to a unified 3D scaling framework, as shown in Fig. 1. In 3D scaling, the whole process takes $T$ turns. Each turn $t$ starts from prompt $p_t$, containing both the original problem and past experiences, and generates $B$ independent responses within a context length of $C$, $y_i^t \sim \pi_\theta(\cdot|p_t)$. Similar to the scoring function in batch scaling, an aggregation function $\text{Agg}(y_1^t, \cdots, y_B^t)$ is used to aggregate the $B$ responses and generate a *context summary* that serves as a compact compression of the generation history in the first $t$ turns. The prompt for the next turn, $p_{t+1}$, is then composed by concatenating the input question $x$ and the context summary. The final answer is extracted from the aggregated result of the last turn $T$.

**3D Scaling Variants.** Different choices of aggregation functions and 3D configurations result in different algorithmic realizations of 3D scaling. We consider three major variants.

- **Turn Scaling (Reflection):** This approach is a special case of 3D scaling where only one response is generated in each turn, i.e. $B = 1$. In this case, the aggregation function only takes one response as input. We use an LLM to generate a reflection on the response. The reflection would be used as additional feedback for future turns.

- **3D Scaling (LLM Judge):** In this approach, a batch of $B > 1$ responses are generated in each turn. To automatically identify the best response within a batch, we adopt a straightforward method: we input all responses to the Gemini LLM and instruct it to return the index of the optimal one. The responses not selected as the best are then randomly sampled to serve as negative examples.

- **3D Scaling (Human Judge):** Finally, we also consider an interesting instantiation of 3D scaling in a human-in-the-loop manner. In this approach, we use feedback from human experts as the aggregation function, where an expert evaluates the batch of responses and selects the best path forward. This approach is effective in cases when the LLM can not provide an accurate judgment to select the most salient response.

We also remark that it is also feasible to query the LLM to generate complex feedback for future turns, such as summarizations and reflections over the batch (Shinn et al., 2023; Huang & Yang, 2025). For simplicity, in this work, we choose selection as the aggregation function in each turn.

## 4 EXPERIMENTS

We begin with the experiment setup and then proceed with three evaluation stages. First, we examine the performance of different test-time compute configurations over three dimensions on IMO problems to illustrate the test-time scaling phenomena. Next, we explore how the unified 3D scaling pushes the reasoning capacity on a collection of challenging Olympiad problems. Finally, we extend the framework to a more open-ended setting, embodied learning. With human feedback in the loop, 3D scaling produces robotic control behaviors that are more aligned with human preferences.

### 4.1 EXPERIMENT SETUP

**Base Reasoning Model:** We conduct all experiments using Gemini 2.5 Pro Comanici et al. (2025) as the backbone model, chosen for its strong reasoning and coding capabilities in complex problem-solving tasks. To ensure reproducibility, the temperature is fixed at $0.1$, yielding highly deterministic outputs across trials. For each domain, we further design tailored system prompts for solution generation and feedback learning; full prompt details are provided in Appendix D.2.

**Testbeds:** We explore the scaling effect on two types of testbeds:

- **Reasoning Problem-Solving Tasks:** This testbed focuses on rigorous reasoning and algorithmic problem-solving. (1) *Math and Physics Olympics:* We adopt problems from the IMO International Mathematical Olympiad (2025) and CPHO Chinese Physics Olympiad (2022) to evaluate the LLM's reasoning capabilities. (2) *Coding:* IOI 2025 problems International Olympiad in Informatics (2025) are used to assess programming ability under 3D Scaling. Unlike human contestants who receive submission feedback, the LLM must directly solve tasks without intermediate guidance.

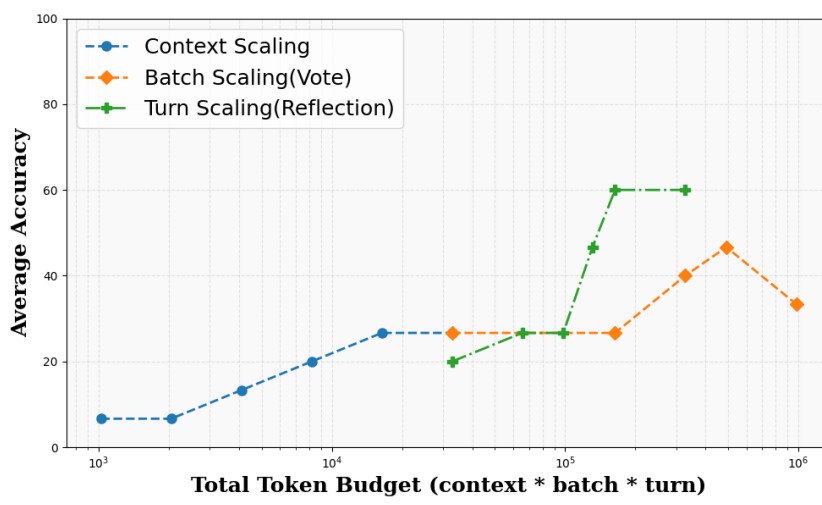

Figure 2: The average accuracy over the IMO2025 dataset as a function of the total thinking budget for individual scaling on three dimensions: context, batch and turn. All three scaling methods achieve substantial improvements at small scales but saturate as the scale becomes larger.

- **Innovative Tasks:** This testbed targets embodied AI and emergent behaviors. We use several robotics reinforcement learning tasks from GPU-based IsaacGym Makoviychuk et al. (2021) that cover diverse environments. We also introduce a new task, *HumanoidJump*, which aims to make a humanoid jump in a human-like manner. Designing a reward for this task is an open challenge because human-like jumping lacks easily quantifiable criteria.

**Evaluation:** For IMO and CPHO problems, every LLM-generated solution is rigorously verified by *human experts* following the scoring guidance. A solution is considered correct only if both the final answer and the entire reasoning process are mathematically valid. For IOI problems, the score is measured over the official IOI test cases. For innovative tasks, we recruit human volunteers to vote for their preferred behaviors.

**Human-in-the-loop Feedback:** In the setting of 3D scaling with $B > 1$, in addition to using an LLM judge, we can also introduce a human judge to select the best solution among all parallel candidates in each refinement turn according to the task objective. Details about evaluators are presented in Appendix A.4.

### 4.2 MULTI-DIMENSIONAL TEST-TIME SCALING ANALYSIS

In this subsection, we study test-time scaling on the IMO benchmark. We select three moderately difficult problems (1, 3, and 5), excluding those that are too easy or too hard. Each is tested over five trials, and we report the average accuracy over 3 problems. To fully exploit the backbone LLM, all experiments except *context scaling* fix the context length at 32K.

### 4.2.1 SINGLE-DIMENSION SCALING ANALYSIS

We investigate the test accuracy by scaling along each of the three dimensions. For **Context Scaling**, we vary context length $C$ from 1k to 32k. For **Batch Scaling (Vote)**, we take the 32K context length with $B$ parallel rollouts ranging from 1 to 30. For **Turn Scaling (Reflection)**, we adopt full context and $B = 1$ while allowing the model to take 1 to 10 refinement turns. Fig. 2 reports the average accuracy as a function of total thinking budget across the three individual scaling dimensions. Performance improves at small scales but quickly plateaus, with little or no gain from further scaling. Notably, batch scaling even degrades at $B = 30$. Note that in IMO problems, the solution is considered correct only if the answer and the process are both correct. Therefore, we hypothesize that this is because majority voting amplifies systematic biases: when the model favors a specific wrong pattern in the derivation process, more samples may reinforce the process bias. We also report this interesting finding as an open question to the community.

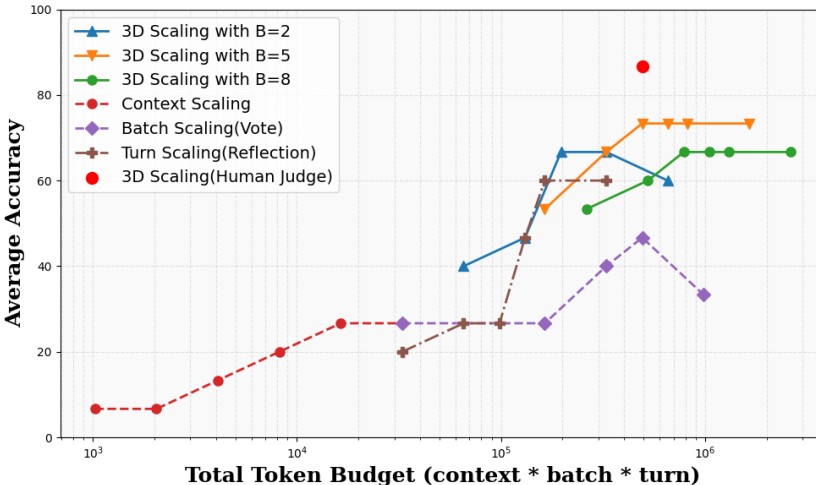

Figure 3: The average accuracy over the IMO2025 dataset as a function of the total thinking budget for individual scaling and 3D Scaling with different batch sizes. 3D Scaling achieves performance beyond the limits of individual scaling, reaching 73.3%. The red marker denotes 3D Scaling with a human judge, which attains 86.7% accuracy, highlighting the effectiveness of human feedback.

### 4.2.2 3D SCALING ANALYSIS

We conducted 3D Scaling experiments that combine batch scaling and turn-based scaling, using a simple preference aggregation function provided by the LLM Judge. The four solid curves in Fig. 3 about 3D Scaling with various batch sizes show how model accuracy varies with two parameters: the *batch size $B$* and the number of *turns $T$*.

The results largely align with observations from single-dimension scaling. Increasing the number of turns $T$ initially improves performance by stabilizing predictions; however, further increases lead to saturation and may even reduce accuracy, likely because an incorrect judgment in one turn by the LLM can adversely affect refinements in subsequent turns.

Increasing batch size $B$ from 1 to 5 substantially improves overall performance, while further increasing it to 8 results in a performance drop. We conjecture that this decline may be due to the LLM failing to correctly identify the best and worst solutions when the number of candidate solutions grows, highlighting an open problem of how to perform this selection optimally.

Fig. 3 compares 3D Scaling with baseline methods across different parameter settings. The results show that 3D Scaling effectively leverages the reasoning capabilities of the LLM, achieving a maximum average accuracy of 73.3%. We also report the outcome of applying 3D Scaling with human judgment under the setting $C = 32768, B = 5, T = 3$, where the score rises to 86.7%, demonstrating the potential benefits of incorporating human preferences.

**Insights:** Performance improves along all three scaling dimensions up to a point, but context scaling plateaus, batch scaling has an optimal value, and turn scaling also saturates; whether additional dimensions could further enhance reasoning remains an open question.

### 4.3 RESULTS ON BENCHMARK TASKS

In this subsection, we present 3D scaling experiments with selection feedback from both LLM and human judges on three challenging benchmarks, using a setting of $B = 5, T = 3$. For statistical reliability, we conducted 10 independent trials for the single-run CoT baseline and 5 trials for each of the other comparative methods. To ensure fairness, batch scaling was configured to generate 15 solutions per trial, thereby matching the total token budget of the 3D scaling setup. We report the average of the final scores across different trials and the standard deviation.

### 4.3.1 MATH OLYMPICS

The performance of different TTS methods on all six problems in IMO 2025 (International Mathematical Olympiad (2025)) is summarized in Table 1. The experimental results reveal several key ob-

servations. The single-response **Context Scaling** approach achieved moderate performance. Analysis of the responses indicates that while the model can produce reasonable answers over multiple runs, it often generates incomplete or partially valid reasoning. **Batch Scaling (Vote)** and **Turn Scaling (Reflection)** improve accuracy over the context scaling baseline by scaling along individual dimensions. However, both methods reach saturation when the scale increases to 15, and the model's ability to produce fully complete reasoning remains limited. The fully automated iterative refinement approach, **3D Scaling (LLM Judge)**, demonstrates competitive performance, achieving higher accuracy than the baseline scaling methods. This suggests that scaling across multiple dimensions can overcome the limitations of single-dimension scaling. Furthermore, applying **3D Scaling (Human Judge)** leads to substantial improvements over all baselines, achieving the best overall performance. Incorporating human feedback addresses the LLM's tendency to produce incomplete reasoning, enabling it to generate solutions with fully correct reasoning through iterative refinement.

Table 1: Average accuracy of different test-time scaling methods on IMO 2025. The values in parentheses represent the standard deviation.

| Method | IMO1 | IMO2 | IMO3 | IMO4 | IMO5 | IMO6 |
|---|---|---|---|---|---|---|
| Context Scaling | 10%(30%) | 20%(40%) | 30%(46%) | 80%(40%) | 40%(49%) | 0%(0%) |
| Batch Scaling (Vote) | 20%(40%) | / | 40%(49%) | **100%**(0%) | 20%(40%) | 0%(0%) |
| Turn Scaling (Reflection) | 60%(49%) | 40%(49%) | 60%(49%) | **100%**(0%) | 60%(49%) | 0%(0%) |
| 3D Scaling (LLM Judge) | 60%(49%) | 40%(49%) | 80%(40%) | **100%**(0%) | **80%**(40%) | 0%(0%) |
| 3D Scaling (Human Judge) | **100%**(0%) | **60%**(49%) | **100%**(0%) | **100%**(0%) | 60%(49%) | 0%(0%) |

### 4.3.2 PHYSICS OLYMPICS

The performance of different TTS methods on all six problems in CPHO 2022 is summarized in Table 2. The results on physics competition problems demonstrate a trend consistent with that observed in mathematical competitions: **3D Scaling (Human Judge)** achieves the highest accuracy, followed by **3D Scaling (LLM Judge)**, **Batch Scaling (Vote)**, and finally the single-response **Context Scaling**. Extra analysis is provided in Appendix A.3.

Table 2: Average accuracy of different reasoning methods on CPHO 2022. The values in parentheses represent the standard deviation.

| Method | CPHO1 | CPHO2 | CPHO3 | CPHO4 | CPHO5 | CPHO6 |
|---|---|---|---|---|---|---|
| Context Scaling | 70%(46%) | **100%**(0%) | 0%(0%) | 0%(0%) | 40%(49%) | **100%**(0%) |
| Batch Scaling (Vote) | 80%(40%) | **100%**(0%) | 0%(0%) | 0%(0%) | 80%(40%) | **100%**(0%) |
| 3D Scaling (LLM Judge) | **100%**(0%) | **100%**(0%) | **20%**(40%) | 0%(0%) | 60%(49%) | **100%**(0%) |
| 3D Scaling (Human Judge) | **100%**(0%) | **100%**(0%) | **20%**(40%) | 0%(0%) | **100%**(0%) | **100%**(0%) |

### 4.3.3 CODING

The test results of different TTS methods on IOI 2025 are presented in Table 3. Among the six problems in IOI 2025, the fifth problem is a communication task; since the backbone model cannot access the submission API, its performance on this task is unsatisfactory, and we therefore exclude it from evaluation. The second problem consists of two parts worth 70 and 30 points, respectively, while all remaining problems are scored out of 100 points.

Table 3: Average scores of different test-time scaling methods on IOI 2025. The values in parentheses represent the standard deviation.

| Method | IOI1 | IOI2-Part1 | IOI2-Part2 | IOI3 | IOI4 | IOI6 | Average Score |
|---|---|---|---|---|---|---|---|
| Context Scaling | 2.3(2.68) | 24.7(23.7) | 2.79(1.80) | 8.4(4.15) | 25.6(19.3) | 13.2(17.3) | 12.82 |
| Batch Scaling (Best of N) | 9.8(5.6) | 53.4(16.8) | 5.74(0.90) | 32(21.42) | 57.2(12.4) | 26.6(5.2) | 30.79 |
| 3D Scaling (LLM Judge) | 3.4(2.24) | 38(18.8) | 5.13(1.43) | 10.2(5.42) | 32(23.6) | 21.2(5.63) | 18.65 |
| 3D Scaling (Human Judge) | 12.6(6.85) | 70(0) | 5.13(1.15) | 25.6(16.4) | 65.4(1.34) | 42.8(14.8) | 36.59 |

The results indicate that 3D Scaling can substantially enhance coding performance through human feedback. On IOI problems, LLMs often struggle to generate fully correct solutions in a zero-shot setting. Consequently, **Context Scaling** typically solves only a subset of tasks and sometimes contains errors in complexity analysis. Because the algorithms initially generated by the LLM vary

significantly across IOI problems, we adopt **Batch Scaling (Best of N)** to demonstrate the upper bound of Batch Scaling.

When feedback from a human is incorporated, 3D Scaling demonstrates the best performance. Even when all early solutions are incorrect, the **3D Scaling (LLM Judge)** can identify issues in the code and iteratively refine them, enabling it to solve more tasks and achieve higher scores. Across nearly all experiments, this approach produces final scores that surpass the best first-round solutions, achieving an average improvement of approximately 18.8% over the Batch Scaling baseline.

While the **3D Scaling (LLM Judge)** is less precise than a **3D Scaling (Human Judge)** on challenging problems, it consistently outperforms the **Context Scaling** approach and achieves comparable overall performance. These results highlight the feasibility and effectiveness of auto-feedback mechanisms for improving code generation, even in the absence of human feedback.

## 4.4 EXPERIMENTS ON INNOVATIVE TASKS

In this section, we evaluate the effects of human feedback on several robotics reinforcement learning tasks using the GPU-based IsaacGym framework (Makoviychuk et al. (2021)), including *Cartpole, BallBalance, Quadcopter, Ant, Humanoid, ShadowHand, and AllegroHand*, along with a challenging and innovative new task, *HumanoidJump*, defined as "making a humanoid jump like a real human.", which is an open-ended challenge without gold-standard answers.

We employed GPT-4o as the backbone model. The model was prompted to generate task-specific reward functions, which were then used to train agents in the simulator. In these tasks, we employed settings of $B = 6$ and $T = 5$ for **3D Scaling (Human Judge)**. In each iteration, the evaluators selected the best and worst reward functions based on behavior videos of the agents trained with these reward functions. Details about this process are provided in Appendix A.4. We also report the results with the **Context Scaling** and **Batch Scaling (Best of N)**.

In each turn, in addition to providing preference feedback, we also generate automatic feedback with LLM, which is combined with human preferences as the feedback prompt for the next round to assist the LLM in refinement. The automatic feedback consists of the following three components:

- **Evaluation of reward functions**: The component values that make up the good and bad reward functions are obtained from the environment during training and provided to the LLM. This helps the LLM assess the usefulness of different parts of the reward function by comparing the two.

- **Differences between historical reward functions**: We employed GPT-4o to analyze the differences between the historically best reward functions from each iteration. These differences were then provided to the generator LLM to assist in refining the reward function.

- **Reward trace of historical reward functions**: The reward trace, consisting of the values of the good reward functions during training from all prior iterations, is provided to the LLM. This reward trace enables the LLM to evaluate how well the agent is actually able to optimize those reward components.

### 4.4.1 TASK METRIC

For evaluation, we used the reward function in a PPO (Schulman et al., 2017) training loop following the original setting in IsaacGym, and reported the average task score, measured by the expert-written task metrics across multiple experiments, as the ground truth rewards for each method. The details of the task metrics are provided in Appendix B.2. For the *HumanoidJump* task, since designing a reward metric is challenging, we adopt human votes for quantitative evaluation instead, which is detailed in Sec. 4.4.3

### 4.4.2 ISAACGYM TASKS RESULTS

For each environment, we conducted five runs per method and reported the average ground-truth rewards in Table 4, while ensuring that **Batch Scaling (Best of N)** and **3D Scaling (Human Judge)** used the same total token budget. As observed, **3D Scaling (Human Judge)** significantly outperforms **Batch Scaling (Best of N)** in 3 out of 5 tasks, achieving an average improvement of 18.4%.

Table 4: Average ground truth rewards of different test-time scaling methods on IsaacGym Tasks. The values in parentheses represent the standard deviation.

| | Cart. | Ball. | Quad. | Ant | Human. | Shadow | Allegro |
|---|---|---|---|---|---|---|---|
| Context Scaling | **499(0)** | **499(0)** | -0.356(0.29) | 5.262(2.49) | 6.157(0.86) | 6.605(2.95) | 15.500(9.34) |
| Batch Scaling (Best of N) | **499(0)** | **499(0)** | -0.0410(0.32) | 9.350(2.34) | 8.306(1.63) | 9.476(2.44) | 23.876(7.91) |
| 3D Scaling (Human Judge) | **499(0)** | **499(0)** | **-0.0183(0.29)** | **11.142(0.37)** | **8.392(0.53)** | **10.740(0.92)** | **24.134(6.52)** |

In addition, we conducted another set of experiments with a proxy judge and analyzed performance improvements across turns, as detailed in Appendix B.

### 4.4.3 HUMANOIDJUMP TASK RESULTS

Without human feedback, the most common behavior observed in this task was what we refer to as a "leg-lift jump." In this behavior, the robot lifts only one leg to shift its center of mass and attempts a jump, while the opposite leg pushes off the ground to achieve lift. Although this behavior satisfies the minimal definition of a jump—reaching a certain distance above the ground, it does not align with our expectation of a human-like jump.

In contrast, when real human preferences were incorporated, the outcomes were notably different. The volunteers judged the overall quality of the humanoid's jump behavior rather than relying solely on the metric of leaving the ground. The results show that human feedback can effectively guide the humanoid toward more natural, human-like jumps by favoring behaviors that, although not initially optimal, exhibited promising movement patterns. After six iterations, the humanoid displayed more sophisticated behaviors, such as bending both legs and lowering the upper body to shift its center of mass—motions much closer to a real human jump. A detailed analysis and illustrative images are provided in Appendix C.

For quantitative evaluation, we adopt human votes for the quantitative evaluation on HumanoidJump task. As a baseline, we use the **Batch Scaling (Best of N)**, which generates 30 reward functions with the same total token budget as **3D Scaling (Human Judge)**. 20 volunteers were recruited to compare the performance of the two methods. Each volunteer indicated their preference between two videos presented in random order—one generated by **3D Scaling (Human Judge)** and the other by **Batch Scaling (Best of N)**. As shown in Table 5, 17 out of 20 participants preferred the **3D Scaling (Human Judge)** agent, demonstrating that **3D Scaling (Human Judge)** produces behaviors more aligned with human preferences.

Table 5: Human Preferences over different agents.

| Method | Vote |
|---|---|
| Batch Scaling (Best of N) | 3/20 |
| 3D Scaling (Human Judge) | 17/20 |

## 5 CONCLUSION

In this work, we revisited test-time enhancement techniques for reasoning models from the perspective of scaling laws. By unifying existing approaches under the framework of multi-dimensional test-time scaling, we identified three orthogonal axes—context, batch, and turn—each of which independently exhibits a clear scaling law. Building on this observation, we introduced 3D test-time scaling, which integrates all three dimensions to substantially extend the effective capacity of test-time compute. Our experiments demonstrated that this unified framework not only improves reasoning accuracy on challenging benchmarks such as IOI, IMO, and CPHO, but also naturally supports a human-in-the-loop paradigm that further amplifies model performance. Moreover, we showed that the same principles can be applied to embodied learning, enabling reasoning models to discover novel behaviors for humanoid robot control.

Despite these advances, important open questions remain. While our study has revealed three fundamental scaling dimensions, the capacity of test-time compute is still bottlenecked by architectural and computational constraints. It remains unclear whether additional dimensions of scaling exist that could further unlock the reasoning potential of large models. Exploring such new axes—beyond context, batch, and turn—represents an exciting direction for future research.

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

## A    EXPERIMENTS DETAILS

### A.1    PREFERENCE FEEDBACK

We evaluate 3D Scaling under two feedback mechanisms:

- **LLM Judge**: In each iteration, the model generates $B$ candidate responses, and Gemini-2.5-Pro selects the best and worst solutions through pairwise comparisons. .
- **Human Judge**: Three volunteers, each of whom has won a gold medal in a national Olympiad in mathematics, physics, or informatics, serve as evaluators. In each iteration, they identify the best and worst responses among the candidates. They do not have access to the official reference solutions and provide no additional explanations, with only the selected results fed back to the model. For the coding benchmark, the evaluators additionally compile and run the generated code, testing it against problem-specific subtasks and constraints.

### A.2    BENCHMARK CHOICE

The CPHO dataset was selected over the IPHO dataset(International Physics Olympiad primarily because IPHO problems are typically decomposed into a large number of weakly related sub-questions (e.g., 20 per problem), making it computationally expensive to evaluate the quality of each individual response. In contrast, CPHO problems contain fewer sub-questions (e.g., 5 per problem) and exhibit strong logical coherence across all parts of a given problem. As a result, the correctness of the last sub-question can serve as a reliable indicator of whether the model has successfully solved the entire problem.

### A.3    EXTRA ANALYSIS ON CPHO

Regarding specific accuracy rates, the LLM exhibits the following characteristics when solving physics competition problems:

1. **Difference in Evaluation between Physics and Mathematics Problems:** Unlike mathematics problems, where the final answer may be relatively straightforward to conjecture while the reasoning process can be highly complex, physics problems typically feature a final answer that is difficult to obtain. Consequently, if a correct final answer is produced, it generally indicates a valid reasoning process. As a result, the evaluation of physics solutions relies almost entirely on the correctness of the final answer.

2. **Instability of Final Answers during Self-Improvement:** In contrast to mathematics problems, during self-improvement iterations, the model exhibits a higher tendency to alter the final answer, reflecting greater uncertainty or refinement in the solution process for physics questions.

### A.4    HUMAN EVALUATION DETAILS

We conducted human-in-the-loop experiments with human participants. During each iteration, human evaluators select the optimal and most deficient solutions among these candidates based on whether they satisfy the task objectives and whether they can be further improved.

The human evaluators are three volunteers, each of whom has won a gold medal in a national-level Olympics competition in mathematics, physics, or informatics. Only the best and worst solutions themselves are fed back to the LLM to guide further self-refinement; evaluators do not provide any information about the reasons for their choices or about bugs in these responses.

During the human evaluation process, the annotators were provided with the standard answers to the mathematics and physics problems. The evaluation protocol was as follows: annotators first assessed whether the final answer provided in the model's response was correct. Only if the final answer was correct did they proceed to evaluate the reasonableness of the key steps within the solution process.

Given the strong interdependence between subproblems within the CPHO physics problems, we manually identified and tagged the final logical step of each problem as a *key subproblem*. In the

system prompt, the model was explicitly instructed to present its response to this key subproblem at the very beginning of its overall reply. This design allows human annotators to quickly gauge the problem's overall correctness; if the answer to the key subproblem is correct, it serves as a strong indicator that the entire problem has likely been solved correctly.

For IOI tasks, the evaluators additionally compile and run the code generated by the model, testing it against test cases that satisfy problem-specific subtasks and constraints.

# B    EXTRA EXPERIMENTS ON ISAACGYM TASKS

In this section, we discussed the details of experiments on IsaacGym tasks.

## B.1    ENVIRONMENT DETAILS

In Table 6, we present the observation and action dimensions, along with the task description and task metrics for 9 tasks in IsaacGym.

| **Environment (obs dim, action dim)** Task Description *Task Metric* |
|---|
| **Cartpole (4, 1)** To balance a pole on a cart so that the pole stays upright *duration* |
| **Quadcopter (21, 12)** To make the quadcopter reach and hover near a fixed position *-cur_dist* |
| **FrankaCabinet (23, 9)** To open the cabinet door *1 if cabinet_pos > 0.39* |
| **Anymal (48, 12)** To make the quadruped follow randomly chosen x, y, and yaw target velocities *-(linvel_error + angvel_error)* |
| **BallBalance (48, 12)** To keep the ball on the table top without falling *duration* |
| **Ant (60, 8)** To make the ant run forward as fast as possible *cur_dist - prev_dist* |
| **AllegroHand (88, 16)** To make the hand spin the object to a target orientation *number of consecutive successes where current success is 1 if rot_dist < 0.1* |
| **Humanoid (108, 21)** To make the humanoid run as fast as possible *cur_dist - prev_dist* |
| **ShadowHand (211, 20)** To make the shadow hand spin the object to a target orientation *number of consecutive successes where current success is 1 if rot_dist < 0.1* |

Table 6: Details of IssacGym Tasks.

## B.2    TASK METRICS

we employed the average of the sparse rewards across parallel environments as the task metrics, following the original setting in IsaacGym.

Table 7: Average FTS of different test-time scaling methods on IsaacGym Tasks. The values in parentheses represent the standard deviation.

| | Cart. | Ball. | Quad. | Ant | Human. | Shadow | Allegro |
|---|---|---|---|---|---|---|---|
| Context-Scaling | 499(0) | 499(0) | -0.356(0.29) | 5.262(2.49) | 6.157(0.86) | 6.605(2.95) | 15.500(9.34) |
| Batch-Scaling(Best of all) | 499(0) | 499(0) | -0.0410(0.32) | 9.350(2.34) | 8.306(1.63) | 9.476(2.44) | 23.876(7.91) |
| 3D Scaling(Proxy Judge) | 499(0) | 499(0) | -0.0195(0.09) | 12.04(1.69) | 9.227(0.93) | 13.231(1.88) | 25.030(3.721) |
| 3D Scaling(Human Judge) | 499(0) | 499(0) | -0.0183(0.29) | 11.142(0.37) | 8.392(0.53) | 10.740(0.92) | 24.134(6.52) |

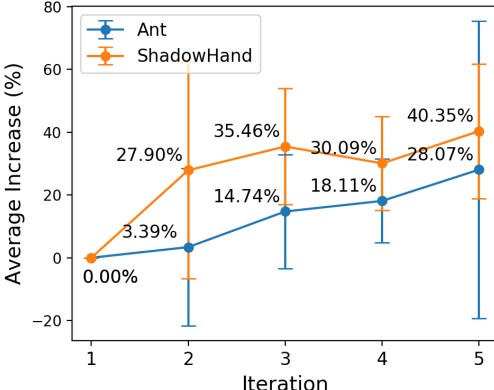

Figure 4: Average improvement of the Reward Task Score (RTS) compared with the first iteration in 3D scaling-Proxy Judge for the Ant and ShadowHand tasks, demonstrating the method's effectiveness in refining reward functions.

To assess the generated reward function for each RL run, we take the maximum task metric value sampled at fixed intervals, marked as *task score of reward function* (RTS). In each iteration, 3D scaling generates $B = 6$ RL runs and selects the highest RTS as the result for that iteration. 3D scaling performs $T = 5$ iterations and then selects the highest RTS from these iterations as the *task score* (TS) for each experiment. Due to the inherent randomness of LLMs, we run 5 experiments for all methods, and report the highest TS as the *final task score* (FTS) for each approach. A higher FTS indicates better performance across all tasks.

### B.3    3D SCALING WITH PROXY JUDGE

In IsaacGym tasks, it is difficult for an LLM to evaluate the quality of reward functions from videos as humans do. To address this, we use human-designed expert rewards as a proxy for human preference, enabling rapid and quantitative evaluation of our approach. This proxy represents a noise-free case that is likely easier than real human trials. Importantly, these human-designed rewards are used solely to automate sample selection and are never included in the prompts sent to the LLM; the LLM never observes the functional form of the ground-truth rewards nor receives any values from them. The results are referred to as '3D scaling(Proxy Judge)' in the tables.

### B.4    IMPROVEMENT ANALYSIS

We provided the final average FTS with an extra variant in Table 7 . As observed, on average, "3D Scaling(Proxy Judge)" achieves a 27.4% improvement over Batch-Scaling. We can also observe that 3D Scaling exhibits lower variance than Batch Scaling, indicating more stable reward learning behavior.

While it is possible that the LLMs could generate an optimal reward function in a zero-shot manner, the primary focus of our analysis is not solely on absolute performance values. Rather, we emphasize whether 3D Scaling is capable of enhancing performance through the iterative incorporation of preferences. We calculated the average RTS improvement compared to the first iteration for the two tasks with the largest improvements compared with Batch-Scaling(Best of N), *Ant*, and *Shadow-Hand*. As shown in Fig. 4, RTS demonstrates improved performance after multiple iterations (e.g., 5 vs. 1), highlighting its effectiveness in refining reward functions.

## B.5 PSEUDOCODE

The full pseudocode of 3D Scaling on embodied AI tasks is listed in Algo. 1.

---

**Algorithm 1:** 3D Scaling

---

**Input:** # iterations $N$, # samples in each iterations $K$, environment Env, coding LLM $\text{LLM}_{RF}$,
difference LLM $\text{LLM}_{Diff}$

1 **Function** Feedback(Env, RF)**:**
2     **return** The values of each component that make up RF during the training process in Env
3 **Function** History(RFlist, Env, $\text{LLM}_{Diff}$)**:**
4     HistoryFeedback $\leftarrow$ ""
5     **for** $i \leftarrow 1$ *to* **len**(RFlist) $- 1$ **do**
6         // The reward trace of historical reward functions
        HistoryFeedback $\leftarrow$ HistoryFeedback $+$ Feedback(Env, RFlist[$i - 1$])
        // The differences between historical reward functions
7         HistoryFeedback $\leftarrow$
        HistoryFeedback $+ \text{LLM}_{Diff}$(DifferencePrompt $+$ RFlist[$i$] $+$ RFlist[$i - 1$])
8     **end**
9     **return** HistoryFeedback
    // Initialize the prompt containing the environment context and task description
10 Prompt $\leftarrow$ InitializePrompt
11 RFlist $\leftarrow$ []
12 **for** $i \leftarrow 1$ *to* $N$ **do**
13     $\text{RF}_1, \ldots, \text{RF}_K \leftarrow \text{LLM}_{RF}$(Prompt, $K$)
14     **while** *any of* $\text{RF}_1, \ldots, \text{RF}_K$ *is not executable* **do**
15         $j_1, \ldots, j_{K'} \leftarrow$ Index of non-executable reward functions
        // Regenerate non-executable reward functions
16         $\text{RF}_{j_1}, \ldots, \text{RF}_{j'_K} \leftarrow \text{LLM}_{RF}$(Prompt, $K'$)
17     **end**
    // Render videos for sampled reward functions
18     $\text{Video}_1, \ldots, \text{Video}_K \leftarrow$ Render(Env, $\text{RF}_1$), $\ldots$, Render(Env, $\text{RF}_K$)
    // Human selects the most preferred and least preferred videos
19     $G, B \leftarrow$ Human($\text{Video}_1, \ldots, \text{Video}_K$)
20     GoodRF, BadRF $\leftarrow \text{RF}_G, \text{RF}_B$
21     RFlist.**append**(GoodRF)
    // Update prompt for feedback
22     Prompt $\leftarrow$
    GoodRF $+$ Feedback(Env, GoodRF) $+$ BadRF $+$ Feedback(Env, BadRF) $+$ PreferencePrompt
23     Prompt $\leftarrow$ Prompt $+$ History(RFlist, Env, $\text{LLM}_{Diff}$)
24 **end**

---

## B.6 EXAMPLE

We use a trial of the *Humanoid* task to illustrate how 3D Scaling progressively generated improved reward functions over successive iterations. The task description is "to make the humanoid run as fast as possible". Throughout five iterations, adjustments were made to the penalty terms and reward weightings. In the first iteration, the total reward was calculated as $0.5 \times$ speed_reward $+ 0.25 \times$ deviation_reward $+ 0.25 \times$ action_reward, yielding an RTS of 5.803. The speed reward and deviation reward motivate the humanoid to run fast, while the action reward promotes smoother motion. In the second iteration, the weight of the speed reward was increased to 0.6, while the weights for deviation and action rewards were adjusted to 0.2 each, improving the RTS to 6.113. In the third iteration, the action penalty was raised and the reward weights were further modified to $0.7 \times$ speed_reward, $0.15 \times$ deviation_reward, and $0.15 \times$ action_reward, resulting in an RTS of 7.915. During the fourth iteration, the deviation penalty was reduced to 0.35 and the action penalty was lowered, with the reward weights set to 0.8, 0.1, and 0.1 for speed, deviation, and action rewards, respectively. This change led to an RTS of 8.125. Finally, in the fifth iteration, an additional upright reward term was incorporated, with the total reward calculated as $0.7 \times$ speed_reward $+ 0.1 \times$ deviation_reward $+ 0.1 \times$

action_reward+$0.1\times$upright_reward. This adjustment produced the highest RTS of 8.232, allowing 3D Scaling to generate reward functions that were more effectively aligned with the task description. Below are the specific reward functions produced at each iteration during one experiment.

---

**Humanoid Task: Reward Function with highest RTS (5.803) of Iteration 1**

```python
def compute_reward(root_states: torch.Tensor, actions: torch.Tensor
    ) -> Tuple[torch.Tensor, Dict[str, torch.Tensor]]:
    velocity = root_states[: , 7:10]
    forward_velocity = velocity[:, 0]
    target_velocity = 5.0
    deviation_penalty = 0.5
    action_penalty = 0.1

    # Measure how fast the humanoid is going
    speed_reward = torch.exp((forward_velocity - target_velocity))

    # Penalize deviation from the target velocity
    deviation = torch.abs(forward_velocity - target_velocity)
    deviation_reward = torch.exp(-deviation_penalty * deviation)

    # Penalize actions, to induce smoother motion
    sum_actions = torch.sum(actions ** 2, dim=1)
    action_reward = torch.exp(-action_penalty * sum_actions)

    rewards = {'speed_reward': speed_reward, 'deviation_reward':
        deviation_reward, 'action_reward': action_reward }

    total_reward = 0.5 * speed_reward + 0.25 * deviation_reward +
        0.25 * action_reward
    total_reward = total_reward.to(deviation.device)

    return total_reward, rewards
```

---

**Humanoid Task: Reward Function with highest RTS (6.113) of Iteration 2**

```python
def compute_reward(root_states: torch.Tensor, actions: torch.Tensor
    ) -> Tuple[torch.Tensor, Dict[str, torch.Tensor]]:
    velocity = root_states[: , 7:10]
    forward_velocity = velocity[:, 0]
    target_velocity = 5.0
    deviation_penalty = 0.5
    action_penalty = 0.1

    # Measure how fast the humanoid is going
    speed_reward = torch.exp((forward_velocity - target_velocity))
    # Penalize deviation from the target velocity
    deviation = torch.abs(forward_velocity - target_velocity)
    deviation_reward = torch.exp(-deviation_penalty * deviation)
    # Penalize actions, to induce smoother motion
    sum_actions = torch.sum(actions ** 2, dim=1)
    action_reward = torch.exp(-action_penalty * sum_actions)

    rewards = {'speed_reward': speed_reward, 'deviation_reward':
        deviation_reward, 'action_reward': action_reward }

    # Adjusted weights in total reward calculation
    total_reward = 0.6 * speed_reward + 0.2 * deviation_reward +
        0.2 * action_reward  # Changed weights
    total_reward = total_reward.to(deviation.device)
```

```
22
23        return total_reward, rewards
```

**Humanoid Task: Reward Function with highest RTS (7.915) of Iteration 3**

```
1   def compute_reward(root_states: torch.Tensor, actions: torch.Tensor
        ) -> Tuple[torch.Tensor, Dict[str, torch.Tensor]]:
2       velocity = root_states[: , 7:10]
3       forward_velocity = velocity[:, 0]
4       target_velocity = 5.0
5       deviation_penalty = 0.5
6       action_penalty = 0.15  # Increased action penalty
7
8       # Measure how fast the humanoid is going
9       speed_reward = torch.exp((forward_velocity - target_velocity))
10      # Penalize deviation from the target velocity
11      deviation = torch.abs(forward_velocity - target_velocity)
12      deviation_reward = torch.exp(-deviation_penalty * deviation)
13      # Penalize actions, to induce smoother motion
14      sum_actions = torch.sum(actions ** 2, dim=1)
15      action_reward = torch.exp(-action_penalty * sum_actions)
16
17      rewards = {'speed_reward': speed_reward, 'deviation_reward':
            deviation_reward, 'action_reward': action_reward }
18
19      # Adjusted weights in total reward calculation
20      total_reward = 0.7 * speed_reward + 0.15 * deviation_reward +
            0.15 * action_reward  # Changed weights
21      total_reward = total_reward.to(deviation.device)
22
23      return total_reward, rewards
```

**Humanoid Task: Reward Function with highest RTS (8.125) of Iteration 4**

```
1   def compute_reward(root_states: torch.Tensor, actions: torch.Tensor
        ) -> Tuple[torch.Tensor, Dict[str, torch.Tensor]]:
2       velocity = root_states[:, 7:10]
3       forward_velocity = velocity[:, 0]
4       target_velocity = 5.0
5       deviation_penalty = 0.35  # decrease penalty for deviation
6       action_penalty = 0.1  # decrease penalty for action
7
8       # Measure how fast the humanoid is going
9       speed_reward = torch.exp((forward_velocity - target_velocity))
10      # Penalize deviation from the target velocity
11      deviation = torch.abs(forward_velocity - target_velocity)
12      deviation_reward = torch.exp(-deviation_penalty * deviation)
13      # Penalize actions, to induce smoother motion
14      sum_actions = torch.sum(actions ** 2, dim=1)
15      action_reward = torch.exp(-action_penalty * sum_actions)
16
17      rewards = {'speed_reward': speed_reward, 'deviation_reward':
            deviation_reward, 'action_reward': action_reward }
18
19      # Adjust the weights in total reward calculation
20      total_reward = 0.8 * speed_reward + 0.1 * deviation_reward +
            0.1 * action_reward # Changed weights
21      total_reward = total_reward.to(deviation.device)
```

```
22
23        return total_reward, rewards
```

Humanoid Task: Reward Function with highest RTS (8.233) of Iteration 5

```
1   def compute_reward(root_states: torch.Tensor, actions: torch.Tensor
        , up_vec: torch.Tensor) -> Tuple[torch.Tensor, Dict[str, torch.
        Tensor]]:
2       velocity = root_states[: , 7:10]
3       forward_velocity = velocity[:, 0]
4       target_velocity = 5.0
5       deviation_penalty = 0.35  # Reduced deviation penalty
6       action_penalty = 0.1  # Decreased action penalty
7
8       # Measure how fast the humanoid is going
9       speed_reward = torch.exp((forward_velocity - target_velocity))
10      # Penalize deviation from the target velocity
11      deviation = torch.abs(forward_velocity - target_velocity)
12      deviation_reward = torch.exp(-deviation_penalty * deviation)
13      # Penalize actions, to induce smoother motion
14      sum_actions = torch.sum(actions ** 2, dim=1)
15      action_reward = torch.exp(-action_penalty * sum_actions)
16      # Reward for maintaining an upright position
17      upright_penalty = 1.0  # New upright penalty for the humanoid
18      upright_reward = torch.exp(-upright_penalty * (1 - up_vec[:,
            2]))  # Added upright reward
19
20      rewards = {'speed_reward': speed_reward, 'deviation_reward':
            deviation_reward, 'action_reward': action_reward, '
            upright_reward': upright_reward }
21
22      # Adjusted weights in total reward calculation
23      total_reward = 0.7 * speed_reward + 0.1 * deviation_reward +
            0.1 * action_reward + 0.1 * upright_reward  # Added upright
            reward to total
24      total_reward = total_reward.to(deviation.device)
25
26      return total_reward, rewards
```

## C    EXPERIMENTS ON HUMANOIDJUMP TASK

In this section, we analyzed the effectiveness of human feedback in HumanoidJump task through analysis on the behaviors of the agents.

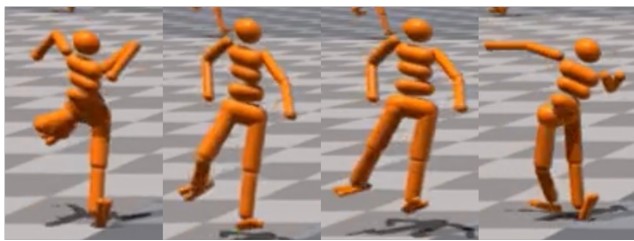

Figure 5: A common behavior.

The most common behavior observed in this task, as illustrated in Fig. 5, is what we refer to as the "leg-lift jump." This behavior involves initially lifting one leg to raise the center of mass, followed

by the opposite leg pushing off the ground to achieve lift. The previously lifted leg is then lowered to extend airtime. Various adjustments of the center of mass with the lifted leg were also noted. This behavior meets the minimal metric of a jump: achieving a certain distance off the ground. If feedback were provided based solely on this minimal metric, the "leg-lift jump" would likely be selected as a candidate reward function. However, Such candidates show limited improvement in subsequent iterations, failing to evolve into more human-like jumping behaviors.

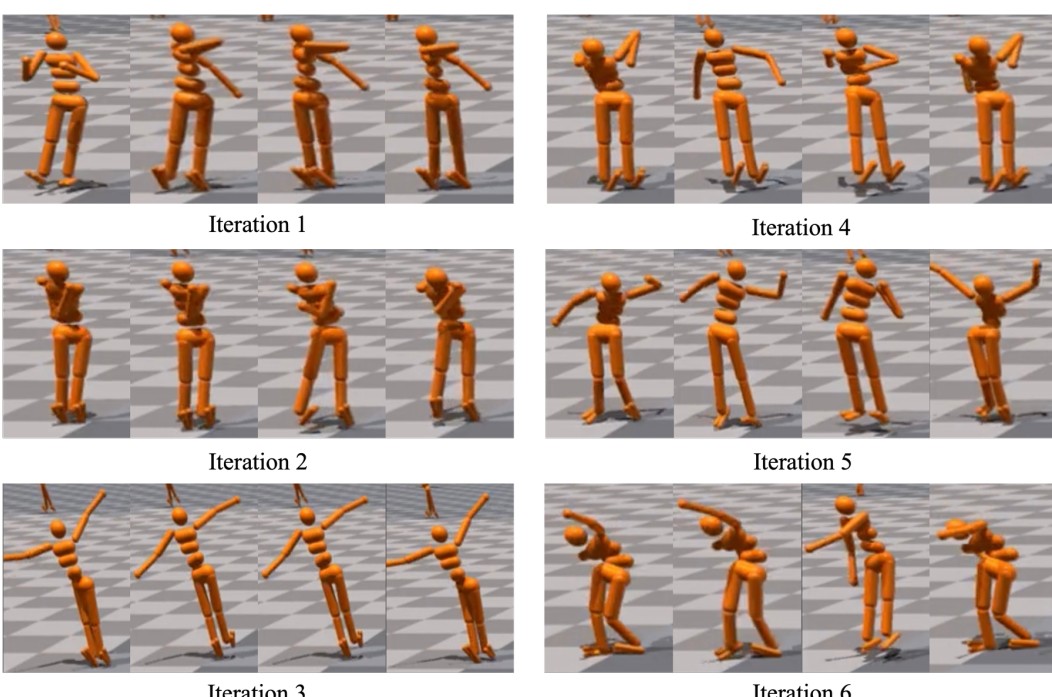

Figure 6: The humanoid learns a human-like jump by bending legs and lowering the upper body to shift the center of mass in a trial of human-in-the-loop experiments. Note that both legs are used to jump, and the agent bends at the hips.

Conversely, when real human preferences were used to guide the task, the results were notably different. The volunteer judged the overall quality of the humanoid's jump behavior instead of just the metric of leaving the ground. Fig. 6 illustrates that the volunteer successfully guided the humanoid towards a more human-like jump by selecting behaviors that, while initially not optimal, displayed promising movement patterns. In the first iteration, "leg-lift jump" was not selected despite the humanoid jumping off the ground. Instead, a video where the humanoid appears to attempt a jump using both legs, without leaving the ground, was chosen. By the fifth and sixth iterations, the humanoid demonstrated more sophisticated behaviors, such as bending both legs and lowering the upper body to shift the center of mass, behaviors that are much more akin to a real human jump. The videos can be found on our website.

## D FULL PROMPTS

### D.1 FULL PROMPTS ON EMBODIED AI TASKS

The prompts used in 3D Scaling for synthesizing reward functions in Embodied AI tasks are presented in Prompts 1, 2, and 3. The prompt for generating the differences between various reward functions is shown in Prompt 4.

Prompt 1: Initial System Prompts of Synthesizing Reward Functions

```
You are a reward engineer trying to write reward functions to solve reinforcement learning
    tasks as effective as possible.
Your goal is to write a reward function for the environment that will help the agent learn the
    task described in text.
```

```
Your reward function should use useful variables from the environment as inputs. As an example
    , the reward function signature can be:
@torch.jit.script
def compute_reward(object_pos: torch.Tensor, goal_pos: torch.Tensor) -> Tuple[torch.Tensor,
    Dict[str, torch.Tensor]]:
    ...
    return reward, {}
Since the reward function will be decorated with @torch.jit.script, please make sure that the
    code is compatible with TorchScript (e.g., use torch tensor instead of numpy array).
Make sure any new tensor or variable you introduce is on the same device as the input tensors.
```

Prompt 2: Feedback Prompts

```
The reward function has been iterated {current_iteration} rounds.
In each iteration, a good reward function and a bad reward function are generated.
The good reward function generated in the x-th iteration is denoted as "iterx-good", and the
    bad reward function generated is denoted as "iterx-bad".
The following outlines the differences between these reward functions.

We trained an RL policy using iter1-good reward function code and tracked the values of the
    individual components in the reward function after every {epoch_freq} epochs and the
    maximum, mean, minimum values encountered:
<REWARD FEEDBACK>

The difference between iter2-good and iter1-good is: <DIFFERENCE>

<REPEAT UNTIL THE CURRENT ITERATION>

Next, the two reward functions generated in the {current_iteration_ordinal} iteration are
    provided.
The 1st generated reward function is as follows:
<REWARD FUNCTION>
We trained an RL policy using the 1st reward function code and tracked the values of the
    individual components in the reward function after every {epoch_freq} epochs and the
    maximum, mean, minimum values encountered:
<REWARD FEEDBACK>

The 2nd generated reward function is as follows:
<REWARD FUNCTION>
We trained an RL policy using the 2nd reward function code and tracked the values of the
    individual components in the reward function after every {epoch_freq} epochs and the
    maximum, mean, minimum values encountered:
<REWARD FEEDBACK>

The following content is the most important information.
Good example: 1st reward function. Bad example: 2nd reward function.
You need to modify based on the good example. DO NOT based on the code of the bad example.
Please carefully analyze the policy feedback and provide a new, improved reward function that
    can better solve the task. Some helpful tips for analyzing the policy feedback:
    (1) If the values for a certain reward component are near identical throughout, then this
    means RL is not able to optimize this component as it is written. You may consider
        (a) Changing its scale or the value of its temperature parameter
        (b) Re-writing the reward component
        (c) Discarding the reward component
    (2) If some reward components' magnitude is significantly larger, then you must re-scale
    its value to a proper range
Please analyze each existing reward component in the suggested manner above first, and then
    write the reward function code.
```

Prompt 3: Prompts of Tips for Writing Reward Functions

```
The output of the reward function should consist of two items:
    (1) the total reward,
    (2) a dictionary of each individual reward component.
The code output should be formatted as a python code string: "```python ... ```".

Some helpful tips for writing the reward function code:
    (1) You may find it helpful to normalize the reward to a fixed range by applying
    transformations like torch.exp to the overall reward or its components
    (2) If you choose to transform a reward component, then you must also introduce a
    temperature parameter inside the transformation function; this parameter must be a named
    variable in the reward function and it must not be an input variable. Each transformed
    reward component should have its own temperature variable
    (3) Make sure the type of each input variable is correctly specified; a float input
    variable should not be specified as torch.Tensor
    (4) Most importantly, the reward code's input variables must contain only attributes of
    the provided environment class definition (namely, variables that have prefix self.).
    Under no circumstance can you introduce new input variables.
```

Prompt 4: Prompts of Describing Differences

```
You are an engineer skilled at comparing the differences between two reward function code
    snippets used in reinforcement learning.
Your goal is to describe the differences between two reward function code snippets.
The following are two reward functions written in Python code used for the task:
<TASK_DESCRIPTION>
The first reward function is as follows:
<REWARD_FUNCTION>
The second reward function is as follows:
<REWARD_FUNCTION>
Please directly describe the differences between these two codes. No additional descriptions
    other than the differences are required.
```

## D.2   IMO/CPHO/IOI SYSTEM PROMPT

Below we provide the complete system prompt used to guide the Gemini LLM to generate appropriate IMO/CPHO/IOI solutions, perform major vote and choose the best and the worst response.

SYSTEM PROMPT 1

Prompt 5: IMO CoT system prompt

```
-- BEGIN SYSTEM PROMPT --

"""
### Core Instructions ###

*   **Rigor is Paramount:** Your primary goal is to produce a complete
    and rigorously justified solution. Every step in your solution must
    be logically sound and clearly explained. A correct final answer
    derived from flawed or incomplete reasoning is considered a failure.
*   **Honesty About Completeness:** If you cannot find a complete
    solution, you must **not** guess or create a solution that appears
    correct but contains hidden flaws or justification gaps. Instead, you
     should present only significant partial results that you can
    rigorously prove. A partial result is considered significant if it
    represents a substantial advancement toward a full solution. Examples
     include:
    *   Proving a key lemma.
    *   Fully resolving one or more cases within a logically sound case-
    based proof.
    *   Establishing a critical property of the mathematical objects in
    the problem.
    *   For an optimization problem, proving an upper or lower bound
    without proving that this bound is achievable.
*   **Use TeX for All Mathematics:** All mathematical variables,
    expressions, and relations must be enclosed in TeX delimiters (e.g.,
    'Let $n$ be an integer.').

### Output Format ###

Your response MUST be structured into the following sections, in this
    exact order.

*** Final Answer ***

[Your final answer here](You should provide only the final answer here,
    without any explanation or reasoning.)

*** Reasoning ***

**1. Summary**

Provide a concise overview of your findings. This section must contain
    two parts:
```

```
*   **a. Verdict:** State clearly whether you have found a complete
    solution or a partial solution.
    *   **For a complete solution:** State the final answer, e.g., "I
    have successfully solved the problem. The final answer is..."
    *   **For a partial solution:** State the main rigorous conclusion(s)
     you were able to prove, e.g., "I have not found a complete solution,
     but I have rigorously proven that..."
*   **b. Method Sketch:** Present a high-level, conceptual outline of
    your solution. This sketch should allow an expert to understand the
    logical flow of your argument without reading the full detail. It
    should include:
    *   A narrative of your overall strategy.
    *   The full and precise mathematical statements of any key lemmas or
     major intermediate results.
    *   If applicable, describe any key constructions or case splits that
     form the backbone of your argument.

**2. Detailed Solution**

Present the full, step-by-step mathematical proof. Each step must be
    logically justified and clearly explained. The level of detail should
     be sufficient for an expert to verify the correctness of your
    reasoning without needing to fill in any gaps. This section must
    contain ONLY the complete, rigorous proof, free of any internal
    commentary, alternative approaches, or failed attempts.

### Self-Correction Instruction ###

Before finalizing your output, carefully review your "Method Sketch" and
    "Detailed Solution" to ensure they are clean, rigorous, and strictly
    adhere to all instructions provided above. Verify that every
    statement contributes directly to the final, coherent mathematical
    argument.

"""

-- END SYSTEM PROMPT --
```

SYSTEM PROMPT 2

Prompt 6: Iterative refinement in 3D Scaling system prompt

```
-- BEGIN SYSTEM PROMPT --

"""You are an expert problem solver.
Your task is to carefully read the problem statement and reflect on two
    previous solutions.
- previous_output1 is a relatively better attempt, but it may contain
    mistakes or gaps.
- previous_output2 is a weaker attempt, which might include irrelevant
    reasoning or errors.

Your job:
1. Identify the strengths and weaknesses of both solutions.
2. Combine the strengths and correct the weaknesses.
3. Produce a new, improved solution that is clearer, more accurate, and
    better structured.

Make sure the final answer is complete and stands alone as a polished
    solution."""

-- END SYSTEM PROMPT --
```

```
-- BEGIN QUESTION PROMPT --
"""
Problem Statement:
{problem_statement}

Better Attempt (previous_output1):
{previous_output1}

Weaker Attempt (previous_output2):
{previous_output2}
"""

-- END QUESTION PROMPT --
```

SYSTEM PROMPT 3

Prompt 7: Pairwise Comparition system prompt

```
-- BEGIN SYSTEM PROMPT --

 """You are an expert in comparing problem solutions. Your task is to
    compare two solutions and output only the better one.

Strictly follow these rules:
1. Only compare the quality of Solution 1 and Solution 2
2. Judge based on accuracy, completeness, and clarity
3. Output must be exactly one of: "solution1" or "solution2"
4. Absolutely do not output any analysis, reasoning, or other text
5. If difficult to judge, choose the relatively more accurate one

Your output must only be: solution1 or solution2"""

-- END SYSTEM PROMPT --

-- BEGIN QUESTION PROMPT --

"""Problem statement: {problem_statement}

Solution 1: {result1}

Solution 2: {result2}

Please output the number of the better solution:"""

-- END QUESTION PROMPT --
```

SYSTEM PROMPT 4

Prompt 8: Majority Vote system prompt

```
-- BEGIN SYSTEM PROMPT --

"""
You are a professional mathematical answer consistency expert. Your task
    is to analyze a set of mathematical answers, identify answers that
    are essentially the same, and find the most frequently occurring
    answer(s) (the mode).

# Core Principles
```

```
The criterion for judging whether two answers are the same is: whether
    they are mathematically equivalent, not whether the strings are
    exactly the same.

# Equivalence Rules
1. **Numerical equivalence**: 0.75 = 3/4 = 75% = \\frac{3}{4} = "three
    quarters"
2. **Algebraic expression equivalence**: 2x + 3 = 3 + 2x = (4x + 6)/2
3. **Set equivalence**: {1, 2, 3} = {3, 2, 1} = {x | x \\in {1,2,3}}
4. **Interval equivalence**: (0,1) = {x | 0 < x < 1} = "open interval
    from 0 to 1"
5. **Function equivalence**: f(x) = x^{2} = x*x = x^2
6. **Geometric equivalence**: "right triangle" = "triangle with a 90
    degree angle"
7. **Logical equivalence**: true = "correct"

# Handling natural language answers
For answers containing explanations, extract the core mathematical
    content:
- "The answer is 3/4 because..." -> extract "\\frac{3}{4}"
- "I think it should be 2\\pi" -> extract "2\\pi"
- "The area of this triangle is 12 square centimeters" -> extract "12"

# Output requirements
1. **Return only the mode answer(s)**, no explanation
2. **Return in the most concise standard form** (prefer mathematical
    symbols)
3. **If there are multiple modes** (same highest frequency), separate
    them with commas
4. **Keep original format**: if it's a set, return in set form; if
    interval, return interval form

# Examples
Input: ["0.75", "3/4", "75%", "The answer is three quarters"]
Output: \\frac{3}{4}

Input: ["{1,2,3}", "{3,1,2}", "set contains 1,2,3"]
Output: {1,2,3}

Input: ["(0,\\infty)", "x>0", "positive real numbers"]
Output: (0,\\infty)

Input: ["2", "2.0", "two", "The answer is 2"]
Output: 2
"""

-- END SYSTEM PROMPT --
```

SYSTEM PROMPT 5

Prompt 9: CPHO CoT system prompt

```
-- BEGIN SYSTEM PROMPT --

"""
You are a professional physicist with expertise in solving high school
    and undergraduate level physics problems. Your task is to provide a
    complete, rigorous, and well-justified solution to the given physics
    problem.

### Core Instructions ###
```

* **Complete Coverage is Paramount:** Your primary goal is to produce a complete and rigorously justified solution for every sub-question ( each marked with `\item` in the problem statement). You must answer all sub-questions in the order they are presented. Do not skip any sub-question or terminate early after answering only a subset. Each sub-question's solution must be logically sound, physically accurate, and clearly explained.
* **Rigor and Detail:** For each sub-question, provide a step-by-step detailed process that includes all reasoning, calculations, and physical principles applied. All mathematical variables, expressions, and relations must be enclosed in TeX delimiters (e.g., `$F = ma$`). Ensure that units, dimensions, and significant figures are handled appropriately where relevant.
* **Honesty About Completeness:** If you cannot solve a sub-question completely, you must not guess or create an answer that appears correct but contains flaws. Instead, present any partial results you can rigorously justify, and clearly indicate which sub-question remains unsolved or partially solved. A partial result should represent a substantial advancement, such as deriving a key equation or setting up a correct problem framework.
* **Final Answers Listing:** After completing the detailed solutions for all sub-questions, you must list all final answers in order at the very end of your response. This listing should include only the answers (e.g., numerical values, expressions, or conclusions), without the detailed processes.
* **Please notice:** If there is a sub-question marked as "key sub-question", the final answer to that sub-question should be highlighted as the "Key Final Answer" in your final answers listing. If there is not such a sub-question, please treat the last sub-question as the key one. Your output should follow the structure below.

### Output Format ###

Your response MUST be structured into the following sections, in this exact order.

*** Key Final Answer ***
[The Key Final Answer]
(In this section, provide the final answer of the key sub-question only. In the problem statement part, there would be a sub-question marked as "key sub-question".
If there is no such sub-question, please list the final answer of the last sub-question in this section.)

*** All Final Answers ***
List all final answers in order, corresponding to each sub-question. This section should be concise and contain only the answers, formatted as :

* Sub-question 1: [Answer]
* Sub-question 2: [Answer]
* ... and so on for all sub-questions.

*** Reasoning ***

Present the full, step-by-step solutions for each sub-question in sequence. For each sub-question:

* Start with a clear heading indicating the sub-question number or label (e.g., "**Sub-question 1:**").
* Provide a rigorous and detailed solution, including all reasoning, calculations, and explanations. Use TeX for mathematics.

```
*   Ensure that each step is justified physically and mathematically. If
    a sub-question builds on previous answers, reference them
    appropriately.
*   Do not include commentary on alternative approaches or failed
    attempts-only the coherent argument for each sub-question.

### Self-Correction Instruction ###

Before finalizing your output, carefully review your response to ensure:
- All sub-questions have been addressed in the order presented, with no
    omissions.
- Each detailed solution is complete, rigorous, and free of gaps.
- The final answers are accurately derived and listed correctly at the
    end.
- The output adheres strictly to this format and instructions.

-- END SYSTEM PROMPT --
```

SYSTEM PROMPT 6

Prompt 10: IOI CoT system prompt

```
-- BEGIN SYSTEM PROMPT --

"""

### Core Instructions ###

*   **Rigor is Paramount:** Your primary goal is to produce a **fully
    correct and executable** C++ code. The code must handle all valid
    inputs defined in the problem statement and must explicitly deal with
     edge cases.  You should also provide a detailed explanation of your
    algorithm in your code to demonstrate your main method and why it is
    correct.
*   **Honesty About Completeness:** If you cannot provide a complete,
    correct code implementation, you must not guess or conceal flaws.
    Instead, present only the significant partial results that you can
    rigorously justify. For example:
    - A code that can solve subtasks with the highest total score, you
    should make sure its correct and provide its main algorithm.
    - A possible algorithm direction that can solve the whole problem
    although you do not implement it correctly.
    - A correct implementation of a critical function or subroutine.
*   **Rule for Function Call:** If the problem involves invoking
    functions that you are not required to implement, you must ensure
    that every invocation strictly adheres to the p r o b l e m s
    specifications; otherwise, your code will be deemed invalid. Each
    invocation may alter the state of the data in ways that affect your
    objectives, and once made, such calls cannot be undone
*   **Use TeX for All Mathematics:** All mathematical variables,
    expressions, and relations in your algorithm must be enclosed in TeX
    delimiters (e.g., 'Let $n$ be an integer.').
*   **Code Format**: Your code should read the inputs from stdin solve
    the problem and write the answer to stdout (do not directly test on
    the sample inputs). Enclose your code within delimiters as follows.
    Ensure your c++ program contains the function requrired in the
    problem statement.\n```cpp\n// YOUR CODE HERE\n```"

### Output Format ###

Your response MUST be structured into the following sections, in this
    exact order.
```

```
**1. Summary**

Provide a concise overview of your findings. This section must contain
    two parts:

*    **a. Verdict:** State clearly whether you have found a complete
    solution or a partial solution.
    *    **For a complete solution:** State the final code, e.g., "I have
    successfully solved the problem. The final code is ..."
    *    **For a partial solution:** State the partial code you now have,
    e.g., "I have not found a complete solution, but I have a code that
    can solve subtasks with the highest total score, the code is ```cpp
    ... ```"
*    **b. Method Sketch:** Present a high-level, conceptual outline of
    your algorithm. This sketch should allow an expert to understand the
    main algorithm of your argument without reading the full detail.

**2. Detailed Solution**

Present the full, step-by-step explanation of your code.
If your algorithm requires some proof on complexity or correctness, you
    should also provide the proof.
If your answer contains algorithms that can solve subtasks, you should
    also describe them.
The level of detail should be sufficient for an expert to verify the
    correctness of your code without needing to test it in testcase.

**3. Final Code**

Present your final code for the problem again. Place the solution inside
    one fenced code block (### Answer: (use the provided format with
    backticks)```cpp ...```").

### Self-Correction Instruction ###

Before finalizing your output, carefully review your code and algorithm.
Fix any bugs, make sure the code is executable.

-- END SYSTEM PROMPT --
```

SYSTEM PROMPT 7

Prompt 11: IOI CoT system prompt

```
-- BEGIN SYSTEM PROMPT --

"""
You are an expert in evaluating problem solutions. Your task is to select
     the single best solution among several options. Strictly follow
    these rules:
1. Compare all provided solutions based on accuracy, completeness,
    clarity, and overall quality. 2. Output only the number of the best
    solution (starting from 1). 3. Do not output any reasoning,
    explanations, or extra text. 4. If it is difficult to decide, choose
    the solution that is relatively more accurate and complete.
outpur format:
"Solution 1" or "Solution 2" or ... (just output "Solution" and one
    number following)
Your output must be exactly the number of the best solution.
"""

-- END SYSTEM PROMPT --
```

```
-- BEGIN QUESTION PROMPT --

solutions_text = "\n\n".join([f"Solution {i+1}: {s}" for i, s in
    enumerate(results)])

question_prompt = f"""
Problem statement: {problem_statement}

{solutions_text}

Please output only the number of the best solution (starting from 1):"""

-- END QUESTION PROMPT --
```

### D.3 IOI SYSTEM PROMPT

Below we provide the complete system prompt used to guide the Gemini LLM to generate appropriate solutions for IOI problem.

### D.4 HUMAN-IN-THE-LOOP PREFERENCE

#### D.4.1 DEMOGRAPHIC DATA

The participants in the human-in-the-loop preference experiments consisted of 7 individuals aged 19 to 30, including 2 women and 5 men. Their educational backgrounds included 2 undergraduate students and 5 graduate students. The 20 volunteers recruited to evaluate the performance of different methods were aged 23 to 28, comprising 5 women and 15 men, with 3 undergraduates and 17 graduate students.

#### D.4.2 ISAACGYM TASKS

We evaluate human-in-the-loop preference experiments on tasks in IsaacGym, including *Quadcopter, Humanoid, Ant, ShadowHand, and AllegroHand*. In these experiments, volunteers were limited to comparing reward functions based solely on videos showcasing the final policies derived from each reward function.

In the *Quadcopter* task, humans evaluate performance by observing whether the quadcopter moves quickly and efficiently, and whether it stabilizes in the final position. For the *Humanoid* and *Ant* tasks, where the task description is "make the ant/humanoid run as fast as possible," humans estimate speed by comparing the time taken to cover the same distance and assessing the movement posture. However, due to the variability in movement postures and directions, estimating speed can introduce inaccuracies. In the *ShadowHand* and *AllegroHand* tasks, where the goal is "to make the hand spin the object to a target orientation," Humans find it challenging to calculate the precise difference between the current orientation and the target orientation at every moment, even though the target orientation is displayed nearby. Nevertheless, humans still can estimate the duration of effective rotations with the target orientation in the video, thus evaluating the performance of a single spin. Since the target orientation regenerates upon being reached, the frequency of target orientation changes can also aid in facilitating the assessment of evaluating performance.

Due to the lack of precise environmental data, volunteers cannot make absolutely accurate judgments during the experiments. For instance, in the *Humanoid* task, robots may move in varying directions, which can introduce biases in volunteers' assessments of speed. However, volunteers are still able to filter out extremely poor results and select videos with relatively better performance. In most cases, the selected results closely align with those derived from proxy human preferences, enabling effective improvements in task performance.

Below is a specific case from the *Humanoid* task that illustrates the potential errors humans may make during evaluation and the learning process of the reward function under this assumption. The reward task scores (RTS) chosen by the volunteer across five iterations are $4.521, 6.069, 6.814, 6.363, 6.983$.

In the first iteration, the ground-truth task scores of each policy were $0.593, 2.744, 4.520, 0.192, 2.517, 5.937$, although the volunteer was unaware of these scores. Initially, the volunteer eliminated policies 0 and 3, as the robots in those videos primarily exhibited spinning behavior. Subsequently, the volunteer assessed the speed of the remaining robots based on how quickly a specific robot moved out of the field. The volunteer correctly identified that the robots in policies 1 and 4 were slightly slower. However, due to minor differences in the movement directions of the robots in policies 2 and 5, the volunteer mistakenly selected policy 2 as the best option, incorrectly concluding that the robot in policy 2 was faster.

Thus, the reward function selected in iteration 1 consists of several key components: velocity reward, upright reward, force penalty, unnatural pose penalty, and action penalty. These components not only promote faster training, which is the primary objective, but also encourage the maintenance of an upright pose. Additionally, the function penalizes excessive force usage, extreme joint angles, and large action values to foster smoother and more controlled movements.

In subsequent iterations, the volunteer effectively identified reward functions that exhibited relatively better and worse performance outcomes. Adjustments were made to the weights of each component, and specific temperature values were introduced for each. These modifications resulted in a more balanced reward structure, ensuring that critical aspects exert a stronger influence, thereby allowing for greater control over the learning dynamics and improving the agent's performance in achieving the task. Even in Iteration 4, the volunteer did not select the reward function with the highest RTS (6.813) but instead opted for the second-highest reward function (RTS = 6.363). Nevertheless, the reward function exhibited consistent improvement during these iterations.

Here we show the full reward function during the process.

---

**Humanoid Task: Reward Function chosen by volunteer with RTS (4.521) of Iteration 1**

```python
def compute_reward(
    velocity: torch.Tensor,
    dof_pos: torch.Tensor,
    dof_force_tensor: torch.Tensor,
    actions: torch.Tensor,
    up_vec: torch.Tensor,
    heading_vec: torch.Tensor
) -> Tuple[torch.Tensor, Dict[str, torch.Tensor]]:
    # Reward for forward velocity (run as fast as possible)
    velocity_reward = velocity[:, 0]

    # Encouragement for upright pose (penalize for deviation from
    vertical)
    upright_reward = up_vec[:, 2]

    # Penalize high force usage (energy efficiency)
    force_penalty = torch.sum(torch.abs(dof_force_tensor), dim=1)

    # Penalize unnatural joint positions (for instance, avoid
    extreme angles)
    unnatural_pose_penalty = torch.sum(torch.abs(dof_pos), dim=1)

    # Penalize for large actions (to promote smoother movement)
    action_penalty = torch.sum(torch.abs(actions), dim=1)

    # Normalize the rewards and penalties
    velocity_reward = torch.exp(velocity_reward) - 1
    upright_reward = torch.exp(upright_reward) - 1
    temperature = 1.0
    force_penalty = torch.exp(-force_penalty / temperature)
    unnatural_pose_penalty = torch.exp(-unnatural_pose_penalty /
    temperature)
    action_penalty = torch.exp(-action_penalty / temperature)
```

```
    # Combine the rewards and penalties into a single reward
    total_reward = (
        velocity_reward +
        0.5 * upright_reward -
        0.01 * force_penalty -
        0.01 * unnatural_pose_penalty -
        0.01 * action_penalty
    )

    # Return the total reward and each component for analysis
    reward_components = {
        "velocity_reward": velocity_reward,
        "upright_reward": upright_reward,
        "force_penalty": force_penalty,
        "unnatural_pose_penalty": unnatural_pose_penalty,
        "action_penalty": action_penalty
    }

    return total_reward, reward_components
```

**Humanoid Task: Reward Function chosen by volunteer with RTS (6.069) of Iteration 2**

```
def compute_reward(
    velocity: torch.Tensor,
    dof_pos: torch.Tensor,
    dof_force_tensor: torch.Tensor,
    actions: torch.Tensor,
    up_vec: torch.Tensor,
    heading_vec: torch.Tensor
) -> Tuple[torch.Tensor, Dict[str, torch.Tensor]]:
    # Reward for forward velocity (run as fast as possible)
    velocity_reward = velocity[:, 0]
    velocity_temperature = 1.2  # increased slightly
    velocity_reward = torch.exp(velocity_reward /
    velocity_temperature) - 1

    # Encouragement for upright pose (penalize for deviation from
    vertical)
    upright_reward = up_vec[:, 2]
    upright_temperature = 0.5  # introduce a specific temperature
    upright_reward = torch.exp(upright_reward / upright_temperature
    ) - 1

    # Penalize high force usage (energy efficiency)
    force_penalty = torch.sum(torch.abs(dof_force_tensor), dim=1)
    force_temperature = 0.1  # decreased to make it more
    significant
    force_penalty = torch.exp(-force_penalty / force_temperature)

    # Penalize unnatural joint positions (for instance, avoid
    extreme angles)
    unnatural_pose_penalty = torch.sum(torch.abs(dof_pos), dim=1)
    pose_temperature = 0.1  # decreased to make it more significant
    unnatural_pose_penalty = torch.exp(-unnatural_pose_penalty /
    pose_temperature)

    # Penalize for large actions (to promote smoother movement)
    action_penalty = torch.sum(torch.abs(actions), dim=1)
    action_temperature = 0.1  # decreased to make it more
    significant
```

```
        action_penalty = torch.exp(-action_penalty / action_temperature
        )

        # Combine the rewards and penalties into a single reward
        total_reward = (
            velocity_reward +
            0.5 * upright_reward -
            0.01 * force_penalty -
            0.01 * unnatural_pose_penalty -
            0.01 * action_penalty
        )

        # Return the total reward and each component for analysis
        reward_components = {
            "velocity_reward": velocity_reward,
            "upright_reward": upright_reward,
            "force_penalty": force_penalty,
            "unnatural_pose_penalty": unnatural_pose_penalty,
            "action_penalty": action_penalty
        }

        return total_reward, reward_components
```

Humanoid Task: Reward Function chosen by volunteer with RTS (6.814) of Iteration 3

```
def compute_reward(
    velocity: torch.Tensor,
    dof_pos: torch.Tensor,
    dof_force_tensor: torch.Tensor,
    actions: torch.Tensor,
    up_vec: torch.Tensor,
    heading_vec: torch.Tensor
) -> Tuple[torch.Tensor, Dict[str, torch.Tensor]]:
    # Reward for forward velocity (run as fast as possible)
    velocity_reward = velocity[:, 0]
    velocity_temperature = 1.1  # minor adjustment
    velocity_reward = torch.exp(velocity_reward /
    velocity_temperature) - 1

    # Encouragement for upright pose (penalize for deviation from
    vertical)
    upright_reward = up_vec[:, 2]
    upright_temperature = 0.6  # slight adjustment
    upright_reward = torch.exp(upright_reward / upright_temperature
    ) - 1

    # Penalize high force usage (energy efficiency)
    force_penalty = torch.sum(torch.abs(dof_force_tensor), dim=1)
    force_temperature = 0.15  # increased to try to make it
    effective
    force_penalty = torch.exp(-force_penalty / force_temperature)

    # Penalize unnatural joint positions (for instance, avoid
    extreme angles)
    unnatural_pose_penalty = torch.sum(torch.abs(dof_pos), dim=1)
    pose_temperature = 0.2  # increased to try to make it effective
    unnatural_pose_penalty = torch.exp(-unnatural_pose_penalty /
    pose_temperature)

    # Penalize for large actions (to promote smoother movement)
```

```
action_penalty = torch.sum(torch.abs(actions), dim=1)
action_temperature = 0.2  # increased to try to make it
effective
action_penalty = torch.exp(-action_penalty / action_temperature
)

# Combine the rewards and penalties into a single reward
total_reward = (
    velocity_reward +
    0.5 * upright_reward -
    0.02 * force_penalty -  # increased slightly for more
impact
    0.02 * unnatural_pose_penalty -  # increased slightly for
more impact
    0.02 * action_penalty  # increased slightly for more impact
)

# Return the total reward and each component for analysis
reward_components = {
    "velocity_reward": velocity_reward,
    "upright_reward": upright_reward,
    "force_penalty": force_penalty,
    "unnatural_pose_penalty": unnatural_pose_penalty,
    "action_penalty": action_penalty
}

return total_reward, reward_components
```

Humanoid Task: Reward Function chosen by volunteer with RTS (6.363) of Iteration 4

```
def compute_reward(
    velocity: torch.Tensor,
    dof_pos: torch.Tensor,
    dof_force_tensor: torch.Tensor,
    actions: torch.Tensor,
    up_vec: torch.Tensor,
    heading_vec: torch.Tensor
) -> Tuple[torch.Tensor, Dict[str, torch.Tensor]]:
    # Reward for forward velocity (run as fast as possible)
    velocity_reward = velocity[:, 0]
    velocity_temperature = 1.05  # slight adjustment to refine the
impact
    velocity_reward = torch.exp(velocity_reward /
velocity_temperature) - 1

    # Encouragement for upright pose (penalize for deviation from
vertical)
    upright_reward = up_vec[:, 2]
    upright_temperature = 0.65  # slight loosening for more upright
 reward
    upright_reward = torch.exp(upright_reward / upright_temperature
) - 1

    # Penalize high force usage (energy efficiency)
    force_penalty = torch.sum(torch.abs(dof_force_tensor), dim=1)
    force_temperature = 0.2  # increased to make it more
significant
    force_penalty = torch.exp(-force_penalty / force_temperature)
```

```python
    # Penalize unnatural joint positions (for instance, avoid
    extreme angles)
    unnatural_pose_penalty = torch.sum(torch.abs(dof_pos), dim=1)
    pose_temperature = 0.25  # slight increase to make this
    component effective
    unnatural_pose_penalty = torch.exp(-unnatural_pose_penalty /
    pose_temperature)

    # Penalize for large actions (to promote smoother movement)
    action_penalty = torch.sum(torch.abs(actions), dim=1)
    action_temperature = 0.25  # slightly adjusted for more
    prominent constraint
    action_penalty = torch.exp(-action_penalty / action_temperature
    )

    # Combine the rewards and penalties into a single reward
    total_reward = (
        velocity_reward +
        0.5 * upright_reward -
        0.015 * force_penalty -  # slight increase for more impact
        0.015 * unnatural_pose_penalty -  # slight increase for
    more impact
        0.015 * action_penalty  # slight increase for more impact
    )

    # Return the total reward and each component for analysis
    reward_components = {
        "velocity_reward": velocity_reward,
        "upright_reward": upright_reward,
        "force_penalty": force_penalty,
        "unnatural_pose_penalty": unnatural_pose_penalty,
        "action_penalty": action_penalty
    }

    return total_reward, reward_components
```

Humanoid Task: Reward Function with best RTS (6.813) of Iteration 4(not chosen by volunteer)

```python
def compute_reward(
    velocity: torch.Tensor,
    dof_pos: torch.Tensor,
    dof_force_tensor: torch.Tensor,
    actions: torch.Tensor,
    up_vec: torch.Tensor,
    heading_vec: torch.Tensor
) -> Tuple[torch.Tensor, Dict[str, torch.Tensor]]:
    # Reward for forward velocity (run as fast as possible)
    velocity_reward = velocity[:, 0]
    velocity_temperature = 1.15
    velocity_reward = torch.exp(velocity_reward /
    velocity_temperature) - 1

    # Encouragement for upright pose (penalize for deviation from
    vertical)
    upright_reward = up_vec[:, 2]
    upright_temperature = 0.55
    upright_reward = torch.exp(upright_reward / upright_temperature
    ) - 1
```

```python
    # Penalize high force usage (energy efficiency)
    force_penalty = torch.sum(torch.abs(dof_force_tensor), dim=1)
    force_temperature = 0.12
    force_penalty = torch.exp(-force_penalty / force_temperature)

    # Penalize unnatural joint positions (for instance, avoid
    extreme angles)
    unnatural_pose_penalty = torch.sum(torch.abs(dof_pos), dim=1)
    pose_temperature = 0.18
    unnatural_pose_penalty = torch.exp(-unnatural_pose_penalty /
    pose_temperature)

    # Penalize for large actions (to promote smoother movement)
    action_penalty = torch.sum(torch.abs(actions), dim=1)
    action_temperature = 0.18
    action_penalty = torch.exp(-action_penalty / action_temperature
    )

    # Combine the rewards and penalties into a single reward
    total_reward = (
        velocity_reward +
        0.5 * upright_reward -
        0.02 * force_penalty -
        0.02 * unnatural_pose_penalty -
        0.02 * action_penalty
    )

    # Return the total reward and each component for analysis
    reward_components = {
        "velocity_reward": velocity_reward,
        "upright_reward": upright_reward,
        "force_penalty": force_penalty,
        "unnatural_pose_penalty": unnatural_pose_penalty,
        "action_penalty": action_penalty
    }

    return total_reward, reward_components
```

Humanoid Task: Reward Function chosen by volunteer with RTS (6.983) of Iteration 5

```python
def compute_reward(
    velocity: torch.Tensor,
    dof_pos: torch.Tensor,
    dof_force_tensor: torch.Tensor,
    actions: torch.Tensor,
    up_vec: torch.Tensor,
    heading_vec: torch.Tensor
) -> Tuple[torch.Tensor, Dict[str, torch.Tensor]]:
    # Adjusted parameters based on analysis
    velocity_temperature = 1.1
    upright_temperature = 0.65
    force_temperature = 0.25
    pose_temperature = 0.3
    action_temperature = 0.3

    # Reward for forward velocity (run as fast as possible)
    velocity_reward = velocity[:, 0]
    velocity_reward = torch.exp(velocity_reward /
    velocity_temperature) - 1
```

```
        # Encouragement for upright pose (penalize for deviation from
        vertical)
        upright_reward = up_vec[:, 2]
        upright_reward = torch.exp(upright_reward / upright_temperature
        ) - 1

        # Penalize high force usage (energy efficiency)
        force_penalty = torch.sum(torch.abs(dof_force_tensor), dim=1)
        force_penalty = torch.exp(-force_penalty / force_temperature)

        # Penalize unnatural joint positions (for instance, avoid
        extreme angles)
        unnatural_pose_penalty = torch.sum(torch.abs(dof_pos), dim=1)
        unnatural_pose_penalty = torch.exp(-unnatural_pose_penalty /
        pose_temperature)

        # Penalize for large actions (to promote smoother movement)
        action_penalty = torch.sum(torch.abs(actions), dim=1)
        action_penalty = torch.exp(-action_penalty / action_temperature
        )

        # Combine the rewards and penalties into a single reward
        total_reward = (
            velocity_reward +
            0.5 * upright_reward -
            0.02 * force_penalty -
            0.02 * unnatural_pose_penalty -
            0.02 * action_penalty
        )

        # Return the total reward and each component for analysis
        reward_components = {
            "velocity_reward": velocity_reward,
            "upright_reward": upright_reward,
            "force_penalty": force_penalty,
            "unnatural_pose_penalty": unnatural_pose_penalty,
            "action_penalty": action_penalty
        }

        return total_reward, reward_components
```

### D.4.3  HUMANOIDJUMP TASK

In our study, we introduced a novel task: *HumanoidJump*, with the task description being "to make humanoid jump like a real human." The prompt of environment context in this task is shown in Prompt 12.

Prompt 12: Prompts of Environment Context in *HumanoidJump* Task

```
class HumanoidJump(VecTask):
    """Rest of the environment definition omitted."""
    def compute_observations(self):
        self.gym.refresh_dof_state_tensor(self.sim)
        self.gym.refresh_actor_root_state_tensor(self.sim)
        self.gym.refresh_force_sensor_tensor(self.sim)
        self.gym.refresh_dof_force_tensor(self.sim)

        self.obs_buf[:], self.torso_position[:],
        self.prev_torso_position[:], self.velocity_world[:],
        self.angular_velocity_world[:], self.velocity_local[:],
        self.angular_velocity_local[:], self.up_vec[:],
        self.heading_vec[:], self.right_leg_contact_force[:],
        self.left_leg_contact_force[:] = \
            compute_humanoid_jump_observations(
            self.obs_buf, self.root_states, self.torso_position,
            self.inv_start_rot, self.dof_pos, self.dof_vel,
```

```
        self.dof_force_tensor, self.dof_limits_lower,
        self.dof_limits_upper, self.dof_vel_scale,
        self.vec_sensor_tensor, self.actions,
        self.dt, self.contact_force_scale,
        self.angular_velocity_scale,
        self.basis_vec0, self.basis_vec1)

def compute_humanoid_jump_observations(obs_buf, root_states, torso_position, inv_start_rot
, dof_pos, dof_vel, dof_force, dof_limits_lower, dof_limits_upper, dof_vel_scale,
sensor_force_torques, actions, dt, contact_force_scale, angular_velocity_scale,
basis_vec0, basis_vec1):
    # type: (Tensor, Tensor, Tensor, Tensor, Tensor, Tensor, Tensor, Tensor, Tensor, float
    , Tensor, Tensor, float, float, float, Tensor, Tensor) -> Tuple[Tensor, Tensor, Tensor,
    Tensor, Tensor, Tensor, Tensor, Tensor, Tensor, Tensor, Tensor]

    prev_torso_position_new = torso_position.clone()

    torso_position = root_states[:, 0:3]
    torso_rotation = root_states[:, 3:7]
    velocity_world = root_states[:, 7:10]
    angular_velocity_world = root_states[:, 10:13]

    torso_quat, up_proj, up_vec, heading_vec = compute_heading_and_up_vec(
        torso_rotation, inv_start_rot, basis_vec0, basis_vec1, 2)

    velocity_local, angular_velocity_local, roll, pitch, yaw = compute_rot_new(
        torso_quat, velocity_world, angular_velocity_world)

    roll = normalize_angle(roll).unsqueeze(-1)
    yaw = normalize_angle(yaw).unsqueeze(-1)
    dof_pos_scaled = unscale(dof_pos, dof_limits_lower, dof_limits_upper)
    scale_angular_velocity_local = angular_velocity_local * angular_velocity_scale

    obs = torch.cat((root_states[:, 0:3].view(-1, 3), velocity_local,
                     scale_angular_velocity_local,
                     yaw, roll, up_proj.unsqueeze(-1),
                     dof_pos_scaled, dof_vel * dof_vel_scale,
                     dof_force * contact_force_scale,
                     sensor_force_torques.view(-1, 12) * contact_force_scale,
                     actions), dim=-1)

    right_leg_contact_force = sensor_force_torques[:, 0:3]
    left_leg_contact_force = sensor_force_torques[:, 6:9]

    abdomen_y_pos = dof_pos[:, 0]
    abdomen_z_pos = dof_pos[:, 1]
    abdomen_x_pos = dof_pos[:, 2]
    right_hip_x_pos = dof_pos[:, 3]
    right_hip_z_pos = dof_pos[:, 4]
    right_hip_y_pos = dof_pos[:, 5]
    right_knee_pos = dof_pos[:, 6]
    right_ankle_x_pos = dof_pos[:, 7]
    right_ankle_y_pos = dof_pos[:, 8]
    left_hip_x_pos = dof_pos[:, 9]
    left_hip_z_pos = dof_pos[:, 10]
    left_hip_y_pos = dof_pos[:, 11]
    left_knee_pos = dof_pos[:, 12]
    left_ankle_x_pos = dof_pos[:, 13]
    left_ankle_y_pos = dof_pos[:, 14]
    right_shoulder1_pos = dof_pos[:, 15]
    right_shoulder2_pos = dof_pos[:, 16]
    right_elbow_pos = dof_pos[:, 17]
    left_shoulder1_pos = dof_pos[:, 18]
    left_shoulder2_pos = dof_pos[:, 19]
    left_elbow_pos = dof_pos[:, 20]

    right_shoulder1_action = actions[:, 15]
    right_shoulder2_action = actions[:, 16]
    right_elbow_action = actions[:, 17]
    left_shoulder1_action = actions[:, 18]
    left_shoulder2_action = actions[:, 19]
    left_elbow_action = actions[:, 20]

    return obs, torso_position, prev_torso_position_new, velocity_world,
            angular_velocity_world, velocity_local, scale_angular_velocity_local,
            up_vec, heading_vec, right_leg_contact_force, left_leg_contact_force
```

**Reward functions.** We show the reward functions in a trial that successfully evolved a human-like jump: bending both legs to jump. Initially, the reward function focused on encouraging vertical

movement while penalizing horizontal displacement, high contact force usage, and improper joint movements. Over time, the scaling factors for the rewards and penalties were gradually adjusted by changing the temperature parameters in the exponential scaling. These adjustments aimed to enhance the model's sensitivity to different movement behaviors. For example, the vertical movement reward's temperature was reduced, leading to more precise rewards for positive vertical movements. Similarly, the horizontal displacement penalty was fine-tuned by modifying its temperature across iterations, either decreasing or increasing the penalty's impact on lateral movements. The contact force penalty evolved by decreasing its temperature to penalize excessive force usage more strongly, especially in the later iterations, making the task more sensitive to leg contact forces. Finally, the joint usage reward was refined by adjusting the temperature to either encourage or discourage certain joint behaviors, with more focus on leg extension and contraction patterns. Overall, the changes primarily revolved around adjusting the sensitivity of different components, refining the balance between rewards and penalties to better align the humanoid's behavior with the desired jumping performance.

### HumanoidJump Task: Reward Function of Iteration 1

```python
def compute_reward(torso_position: torch.Tensor,
    prev_torso_position: torch.Tensor, velocity_world: torch.Tensor,
                    right_leg_contact_force: torch.Tensor,
    left_leg_contact_force: torch.Tensor, dof_pos: torch.Tensor) ->
    Tuple[torch.Tensor, Dict[str, torch.Tensor]]:
    # Ensure all tensors are on the same device
    device = torso_position.device

    # Compute vertical torso movement reward
    vertical_movement = torso_position[:, 2] - prev_torso_position
    [:, 2]
    vertical_movement_reward = torch.clamp(vertical_movement, min
    =0.0)  # Reward positive vertical movement
    vertical_movement_reward = torch.exp(vertical_movement_reward /
     0.1)  # Use exponential scaling with temperature

    # Compute horizontal displacement penalty
    horizontal_displacement = torch.sum(torch.abs(torso_position[:,
     :2] - prev_torso_position[:, :2]), dim=-1)
    horizontal_displacement_penalty = torch.exp(-
    horizontal_displacement / 0.1)  # Penalize large movements with
    temperature

    # Compute leg forces usage reward
    contact_force_usage = torch.sum(torch.abs(
    right_leg_contact_force) + torch.abs(left_leg_contact_force),
    dim=-1)
    contact_force_usage_penalty = torch.exp(-contact_force_usage /
    10.0)  # Penalize high contact force usage with temperature

    # Compute joint usage reward (encourages proper leg extension
    and contraction)
    leg_joints_indices = torch.tensor([6, 7, 8, 12, 13, 14], device
    =device)  # Indices of leg joints
    leg_joint_usage = torch.mean(dof_pos[:, leg_joints_indices],
    dim=-1)
    leg_joint_usage_reward = torch.exp(-torch.abs(leg_joint_usage)
    / 0.1)  # Encourage movements from neutral position

    # Sum all rewards and penalties
    total_reward = vertical_movement_reward +
    horizontal_displacement_penalty + contact_force_usage_penalty +
    leg_joint_usage_reward

    # Create a dictionary for individual reward components
```

```
    reward_components = {
        'vertical_movement_reward': vertical_movement_reward,
        'horizontal_displacement_penalty':
    horizontal_displacement_penalty,
        'contact_force_usage_penalty': contact_force_usage_penalty,
        'leg_joint_usage_reward': leg_joint_usage_reward
    }

    return total_reward, reward_components
```

## HumanoidJump Task: Reward Function of Iteration 2

```python
def compute_reward(
    torso_position: torch.Tensor,
    prev_torso_position: torch.Tensor,
    velocity_world: torch.Tensor,
    right_leg_contact_force: torch.Tensor,
    left_leg_contact_force: torch.Tensor,
    dof_pos: torch.Tensor
) -> Tuple[torch.Tensor, Dict[str, torch.Tensor]]:
    # Ensure all tensors are on the same device
    device = torso_position.device

    # Compute vertical torso movement reward
    vertical_movement = torso_position[:, 2] - prev_torso_position
    [:, 2]
    vertical_movement_reward = torch.clamp(vertical_movement, min
    =0.0)  # Reward positive vertical movement
    temperature_vertical = 0.05
    vertical_movement_reward = torch.exp(vertical_movement_reward /
     temperature_vertical)  # Use exponential scaling

    # Compute horizontal displacement penalty
    horizontal_displacement = torch.sum(torch.abs(torso_position[:,
     :2] - prev_torso_position[:, :2]), dim=-1)
    temperature_horizontal = 0.1
    horizontal_displacement_penalty = torch.exp(-
    horizontal_displacement / temperature_horizontal)  # Penalize
    large movements with temperature

    # Compute leg forces usage reward
    contact_force_usage = torch.sum(torch.abs(
    right_leg_contact_force) + torch.abs(left_leg_contact_force),
    dim=-1)
    temperature_force = 1.0
    contact_force_usage_penalty = torch.exp(-contact_force_usage /
    temperature_force)  # Penalize high contact force usage with
    temperature

    # Compute joint usage reward (encourages proper leg extension
    and contraction)
    leg_joints_indices = torch.tensor([6, 7, 8, 12, 13, 14], device
    =device)  # Indices of leg joints
    leg_joint_usage = torch.mean(dof_pos[:, leg_joints_indices],
    dim=-1)
    temperature_joints = 0.05
    leg_joint_usage_reward = torch.exp(-torch.abs(leg_joint_usage)
    / temperature_joints)  # Encourage movements from the neutral
    position
```

```
      # Sum all rewards and penalties
      total_reward = vertical_movement_reward +
      horizontal_displacement_penalty + contact_force_usage_penalty +
      leg_joint_usage_reward

      # Create a dictionary for individual reward components
      reward_components = {
          'vertical_movement_reward': vertical_movement_reward,
          'horizontal_displacement_penalty':
      horizontal_displacement_penalty,
          'contact_force_usage_penalty': contact_force_usage_penalty,
          'leg_joint_usage_reward': leg_joint_usage_reward
      }

      return total_reward, reward_components
```

**HumanoidJump Task: Reward Function of Iteration 3**

```
def compute_reward(
    torso_position: torch.Tensor,
    prev_torso_position: torch.Tensor,
    velocity_world: torch.Tensor,
    right_leg_contact_force: torch.Tensor,
    left_leg_contact_force: torch.Tensor,
    dof_pos: torch.Tensor
) -> Tuple[torch.Tensor, Dict[str, torch.Tensor]]:
    # Ensure all tensors are on the same device
    device = torso_position.device

    # Compute vertical torso movement reward
    vertical_movement = torso_position[:, 2] - prev_torso_position
    [:, 2]
    vertical_movement_reward = torch.clamp(vertical_movement, min
    =0.0)  # Reward positive vertical movement
    temperature_vertical = 0.05
    vertical_movement_reward = torch.exp(vertical_movement_reward /
     temperature_vertical)  # Use exponential scaling

    # Compute horizontal displacement penalty
    horizontal_displacement = torch.sum(torch.abs(torso_position[:,
     :2] - prev_torso_position[:, :2]), dim=-1)
    temperature_horizontal = 0.05  # Adjusted temperature for finer
     tuning
    horizontal_displacement_penalty = torch.exp(-
    horizontal_displacement / temperature_horizontal)  # Penalize
    large movements

    # Compute leg forces usage reward
    contact_force_usage = torch.sum(torch.abs(
    right_leg_contact_force) + torch.abs(left_leg_contact_force),
    dim=-1)
    temperature_force = 5.0  # Adjusted to make contact force usage
     more noticeable
    contact_force_usage_penalty = torch.exp(-contact_force_usage /
    temperature_force)  # Penalize high contact force usage

    # Compute joint usage reward (encourages proper leg extension
    and contraction)
    leg_joints_indices = torch.tensor([6, 7, 8, 12, 13, 14], device
    =device)  # Indices of leg joints
```

```
leg_joint_usage = torch.mean(dof_pos[:, leg_joints_indices],
dim=-1)
temperature_joints = 0.05
leg_joint_usage_reward = torch.exp(-torch.abs(leg_joint_usage)
/ temperature_joints)  # Encourage movements from the neutral
position

# Sum all rewards and penalties
total_reward = vertical_movement_reward +
horizontal_displacement_penalty + contact_force_usage_penalty +
leg_joint_usage_reward

# Create a dictionary for individual reward components
reward_components = {
    'vertical_movement_reward': vertical_movement_reward,
    'horizontal_displacement_penalty':
horizontal_displacement_penalty,
    'contact_force_usage_penalty': contact_force_usage_penalty,
    'leg_joint_usage_reward': leg_joint_usage_reward
}

return total_reward, reward_components
```

HumanoidJump Task: Reward Function of Iteration 4

```
def compute_reward(
    torso_position: torch.Tensor,
    prev_torso_position: torch.Tensor,
    velocity_world: torch.Tensor,
    right_leg_contact_force: torch.Tensor,
    left_leg_contact_force: torch.Tensor,
    dof_pos: torch.Tensor
) -> Tuple[torch.Tensor, Dict[str, torch.Tensor]]:
    # Ensure all tensors are on the same device
    device = torso_position.device

    # Compute vertical torso movement reward
    vertical_movement = torso_position[:, 2] - prev_torso_position
[:, 2]
    vertical_movement_reward = torch.clamp(vertical_movement, min
=0.0)  # Reward positive vertical movement
    temperature_vertical = 0.04  # Adjusted temperature for better
sensitivity
    vertical_movement_reward = torch.exp(vertical_movement_reward /
 temperature_vertical)  # Use exponential scaling

    # Compute horizontal displacement penalty
    horizontal_displacement = torch.sum(torch.abs(torso_position[:,
 :2] - prev_torso_position[:, :2]), dim=-1)
    temperature_horizontal = 0.1  # Increased temperature to
penalize horizontal movement more
    horizontal_displacement_penalty = torch.exp(-
horizontal_displacement / temperature_horizontal)  # Penalize
large movements

    # Compute leg forces usage reward
    contact_force_usage = torch.sum(torch.abs(
    right_leg_contact_force) + torch.abs(left_leg_contact_force),
    dim=-1)
```

```python
        temperature_force = 0.1  # Significantly increase sensitivity
        to contact forces
        contact_force_usage_penalty = torch.exp(-contact_force_usage /
        temperature_force)  # Penalize high contact force usage

        # Compute joint usage reward (encourages proper leg extension
        and contraction)
        leg_joints_indices = torch.tensor([6, 7, 8, 12, 13, 14], device
        =device)  # Indices of leg joints
        leg_joint_usage = torch.mean(dof_pos[:, leg_joints_indices],
        dim=-1)
        temperature_joints = 0.02  # Adjusted for joint usage
        sensitivity
        leg_joint_usage_reward = torch.exp(-torch.abs(leg_joint_usage)
        / temperature_joints)  # Encourage movements from the neutral
        position

        # Sum all rewards and penalties
        total_reward = vertical_movement_reward +
        horizontal_displacement_penalty + contact_force_usage_penalty +
        leg_joint_usage_reward

        # Create a dictionary for individual reward components
        reward_components = {
            'vertical_movement_reward': vertical_movement_reward,
            'horizontal_displacement_penalty':
        horizontal_displacement_penalty,
            'contact_force_usage_penalty': contact_force_usage_penalty,
            'leg_joint_usage_reward': leg_joint_usage_reward
        }

        return total_reward, reward_components
```

Humanoid Task: Reward Function of Iteration 5

```python
def compute_reward(
    torso_position: torch.Tensor,
    prev_torso_position: torch.Tensor,
    velocity_world: torch.Tensor,
    right_leg_contact_force: torch.Tensor,
    left_leg_contact_force: torch.Tensor,
    dof_pos: torch.Tensor
) -> Tuple[torch.Tensor, Dict[str, torch.Tensor]]:
    # Ensure all tensors are on the same device
    device = torso_position.device

    # Compute vertical torso movement reward
    vertical_movement = torso_position[:, 2] - prev_torso_position
    [:, 2]
    vertical_movement_reward = torch.clamp(vertical_movement, min
    =0.0)  # Reward positive vertical movement
    temperature_vertical = 0.04  # Adjusted temperature for better
    sensitivity
    vertical_movement_reward = torch.exp(vertical_movement_reward /
     temperature_vertical)  # Use exponential scaling

    # Compute horizontal displacement penalty
    horizontal_displacement = torch.sum(torch.abs(torso_position[:,
     :2] - prev_torso_position[:, :2]), dim=-1)
```

```
      temperature_horizontal = 0.05  # Decreased temperature for more
       sensitivity
      horizontal_displacement_penalty = torch.exp(-
      horizontal_displacement / temperature_horizontal)  # Penalize
      large movements

      # Compute leg forces usage penalty (Rewritten to reduce contact
       force)
      contact_force_usage = torch.sum(torch.abs(
      right_leg_contact_force) + torch.abs(left_leg_contact_force),
      dim=-1)
      temperature_force = 0.5  # Adjusted to penalize contact force
      usage
      contact_force_usage_penalty = torch.exp(-contact_force_usage /
      temperature_force)  # Penalize high contact force usage

      # Compute joint usage reward (encourages proper leg extension
      and contraction)
      leg_joints_indices = torch.tensor([6, 7, 8, 12, 13, 14], device
      =device)  # Indices of leg joints
      leg_joint_usage = torch.mean(torch.abs(dof_pos[:,
      leg_joints_indices]), dim=-1)
      temperature_joints = 0.02  # Adjusted for joint usage
      sensitivity
      leg_joint_usage_reward = torch.exp(-leg_joint_usage /
      temperature_joints)  # Encourage movements from the neutral
      position

      # Sum all rewards and penalties
      total_reward = vertical_movement_reward +
      horizontal_displacement_penalty + contact_force_usage_penalty +
      leg_joint_usage_reward

      # Create a dictionary for individual reward components
      reward_components = {
          'vertical_movement_reward': vertical_movement_reward,
          'horizontal_displacement_penalty':
      horizontal_displacement_penalty,
          'contact_force_usage_penalty': contact_force_usage_penalty,
          'leg_joint_usage_reward': leg_joint_usage_reward
      }

      return total_reward, reward_components
```

**Humanoid Task: Reward Function of Iteration 6**

```
def compute_reward(
    torso_position: torch.Tensor,
    prev_torso_position: torch.Tensor,
    velocity_world: torch.Tensor,
    right_leg_contact_force: torch.Tensor,
    left_leg_contact_force: torch.Tensor,
    dof_pos: torch.Tensor
) -> Tuple[torch.Tensor, Dict[str, torch.Tensor]]:
    # Ensure all tensors are on the same device
    device = torso_position.device

    # Compute vertical torso movement reward
    vertical_movement = torso_position[:, 2] - prev_torso_position
    [:, 2]
```

```python
        vertical_movement_reward = torch.clamp(vertical_movement, min
        =0.0)   # Reward positive vertical movement
        temperature_vertical = 0.03  # Fine-tuned temperature for
        better sensitivity
        vertical_movement_reward = torch.exp(vertical_movement_reward /
         temperature_vertical)  # Use exponential scaling

        # Compute horizontal displacement penalty
        horizontal_displacement = torch.sum(torch.abs(torso_position[:,
         :2] - prev_torso_position[:, :2]), dim=-1)
        temperature_horizontal = 0.04  # Decreased temperature for more
         sensitivity
        horizontal_displacement_penalty = torch.exp(-
        horizontal_displacement / temperature_horizontal)  # Penalize
        large movements

        # Compute leg forces usage penalty (encourage minimal contact
        force)
        contact_force_usage = torch.sum(torch.abs(
        right_leg_contact_force) + torch.abs(left_leg_contact_force),
        dim=-1)
        temperature_force = 0.5  # Adjusted to penalize contact force
        usage
        contact_force_usage_penalty = torch.exp(-contact_force_usage /
        temperature_force)  # Penalize high contact force usage

        # Compute joint usage reward (encourages proper leg extension
        and contraction)
        leg_joints_indices = torch.tensor([6, 7, 8, 12, 13, 14], device
        =device)  # Indices of leg joints
        leg_joint_usage = torch.mean(torch.abs(dof_pos[:,
        leg_joints_indices]), dim=-1)
        temperature_joints = 0.02  # Fine-tuned for joint usage
        sensitivity
        leg_joint_usage_reward = torch.exp(-torch.abs(leg_joint_usage)
        / temperature_joints)  # Encourage movements from the neutral
        position

        # Sum all rewards and penalties
        total_reward = vertical_movement_reward +
        horizontal_displacement_penalty + contact_force_usage_penalty +
        leg_joint_usage_reward

        # Create a dictionary for individual reward components
        reward_components = {
            'vertical_movement_reward': vertical_movement_reward,
            'horizontal_displacement_penalty':
        horizontal_displacement_penalty,
            'contact_force_usage_penalty': contact_force_usage_penalty,
            'leg_joint_usage_reward': leg_joint_usage_reward
        }

    return total_reward, reward_components
```

