# OpenReview forum: "Extending Test-Time Scaling: A 3D Perspective with Context, Batch, and Turn"
_ICLR.cc/2026/Conference — Submitted to ICLR 2026_

### Official Review · Reviewer_WdNs · 2025-10-26

**Soundness:** 2
**Presentation:** 2
**Contribution:** 2
**Rating:** 2
**Confidence:** 3

**Summary:**

This submission introduces 3D test-time scaling, an integration of existing test-time enhancement techniques in three dimensions: context, batch, and turn. Results show that the proposed method improves the performance over multiple benchmarks, such as IOI, IMO, and CPHO. The proposed method also naturally supports human-in-the-loop settings to further amplify the performance.

**Strengths:**

- The submission is well-written and easy to follow.

**Weaknesses:**

- The proposed new method is trivial. It is just a combination of existing test-time scaling method. What is the uniqueness?
- The analysis of the 3D test-time scaling (sec 4.2) is insufficient to provide deeper understanding. For example, what's the reason for the performance plateau and even degradation after scaling?
- Experiment results in Sec. 4.3 are not compelling enough. For example, both the Batch Scaling and Turn Scaling baselines can achieve perfect performance as the proposed method in IMO4, while the proposed method achieves 0 in IMO6. Similarly, the Batch Scaling baseline even achieves better performance than the proposed method with LLM Judge in IOI.
- Results on the IsaacGym tasks are not differentiable if we count the variance. And these tasks seem not meaningful because their human-written reward functions must be used to train the base models.

**Questions:**

See above.

---

### Official Review · Reviewer_SrUw · 2025-10-27

**Soundness:** 2
**Presentation:** 3
**Contribution:** 2
**Rating:** 4
**Confidence:** 3

**Summary:**

This work revisits test-time enhancement techniques through the lens of scaling effect and introduces a unified framework of multi-dimensional test-time scaling to extend the capacity of test-time reasoning. The experimental results show how the concatenation or each dimension of test-time scaling methods can impact the reasoning performance.

**Strengths:**

1. The paper writing is well-structured with clear motivation. The related work makes it easy for the reader to understand the context.

2. The proposed framework was evaluated in multiple benchmarks in both reasoning tasks and embodied AI tasks.

3. The experimental results provide interesting and useful insights regarding how different dimensions of test-time scaling can extend the capacity of test-time reasoning.

**Weaknesses:**

1. The contribution of this work is more like empirical tests about three existing (context, batch, turn) approaches' performance along with the performance when combing the three methods together. There is no fundamental breakthough in model design or algorithm side. The proposed framework seems to be a simple concatenation of existing approaches together.

2. Human evaluation was the key to the experimental results. However, lots of details about human evaluation design were missing. During evaluation, the authors mentioned that "every LLM-generated solution is rigorously verified by human experts following the scoring guidance". How were such 'human experts' recruited and how to prove such expertise? How to ensure that the scoring process is fair across different models and different questions? Are there any repeated-measure design to rate each response by different experts and aggregate the scores together?

  Similarly, for innovative tasks, the authors stated that "we recruit human volunteers to vote for their preferred behaviors". What are the voting criteria? How were volunteers recruited and how many volunteers were responsible for each response? Did the human volunteers sign the consent forms and was this human-subject study approved by the local institute? I found one statement in line 1006: "The volunteer judged the overall quality of the humanoid’s jump behavior instead of just the metric of leaving the ground". This statement itself looks confusing. And how was such "overall quality" measured?

  Another statement in line 1007 was that "the volunteer successfully guided the humanoid towards a more human-like jump by selecting behaviors that, while initially not optimal, displayed promising movement patterns". Does this mean that the volunteer serves as both human judge and human evaluator? And is there only one volunteer in the embodied experiment of Gym?

3. It seems that the authors mixed the human judges (Appendix A1) and human evaluators (Appendix A4) together. Here the human judges were used to provide feedback to improve LLMs' problem solving, while human evaluators were used to evaluate different models, including the one model with human judges' feedback. If human judges and human evaluators were the same participants, then the major concern is that such evaluation is not fair and can probably introduce bias into results.

4. The testing sample was limited in Section 4.2, i.e. only three problems were tested and each was tested over five trials.

5. It is not surprising that involving human judges can result in better performance, especially when the recruited human judges are experts in Olympiad, as depicted in line 602-603. Involving human judges into LLMs' problem solving itself is not a fair comparison. Although human judges did not provide any information about the reasons for their choices, providing best and worst solutions as feedback to the LLM was probably enough to guide further self-refinement. This is not surprising as well.


6. The goal of this work is ambitious, i.e., extending the capacity of test-time scaling. However, the exploration and testing only used one language model (Gemini 2.5 Pro), which significantly reduced the rigor, significance, and generalization of this work.

**Questions:**

Please check my concerns and questions in Weaknesses section.

**Details Of Ethics Concerns:**

No information about human subject recruitment procedure. How were human subjects recruited? Did the human subjects sign the consent forms and was this human-subject study approved by the local institute?

---

### Official Review · Reviewer_ULYS · 2025-10-30

**Soundness:** 3
**Presentation:** 3
**Contribution:** 2
**Rating:** 6
**Confidence:** 4

**Summary:**

The paper proposes a unified framework for **test-time scaling (TTS)** across three axes: **Context (C)**, **Batch (B)**, and **Turn (T)**, and instantiates *"3D scaling"* methods that combine these axes via an aggregation function (LLM or human judge).

Formally, *context scaling* truncates generation to a token budget *C*; *batch scaling* samples *B* candidates and selects one by a scoring function or LLM; *turn scaling* iteratively refines across *T* turns by concatenating a "context summary" into the next prompt.

The authors evaluate on **IMO 2025**, **CPHO 2022**, and **IOI 2025**, and further attempt an embodied RL case study (**IsaacGym**) where the LLM designs reward functions iteratively with human preferences.

**Strengths:**

- **Clear formalization of three TTS axes (C, B, T):**
  The paper presents a unified 3D test-time scaling (TTS) procedure with aggregation, including both LLM- and human-judge variants. This provides practitioners with a coherent way to reason about *length* ($C$), *breadth* ($B$), and *depth* ($T$) at inference.

- **Empirical breadth across domains:**
  The evaluation spans mathematics, physics, and code, as well as embodied RL reward design. This cross-domain scope is valuable, even if the methodological rigor varies.

- **Human-in-the-loop aggregation:**
  The paper operationalizes expert selection as an aggregation function. Human selection often outperforms the LLM-judge, aligning with practitioner experience.

- **Transparency in limitations:**
  The authors clearly discuss saturation and failure modes. They note that batch scaling can degrade at large $B$, and that turn scaling can propagate errors through judge decisions.

- **Creative embodied RL case study:**
  The IsaacGym experiment is innovative. Algorithm 1 is well-described, and the design of three automatic feedback channels (component values, differences between historical bests, and reward traces) reflects a thoughtful blend of numeric and textual feedback.

**Weaknesses:**

- **Ambiguous compute accounting:**
  The "total thinking budget" is defined as the theoretical maximum token count, yet 3D methods introduce additional LLM calls (judge, reflection) and expanding prompts across turns. These overheads are not clearly accounted for, making Figure 3 comparisons unreliable and potentially attributing gains to untracked compute. The authors should explicitly include judge tokens and prompt growth on the x-axis and report compute-matched baselines.

- **Small, high-variance testbeds with limited statistical reporting:**
  The IMO/CPHO evaluations use only six problems each, and reported standard deviations are large. No confidence intervals or statistical tests are provided in the figures, making claims such as “3D scaling extends test-time scaling” statistically fragile. The paper should include confidence intervals, more problems, or use public benchmarks such as GSM8K-hard, MATH-500, or AIME-like datasets.

- **Inconsistent or weak aggregator definitions and baselines:**
  The Batch (Vote) aggregator is reasonable for numeric outputs but not meaningful for derivational physics or math. For math, process correctness is required; for physics, the metric reduces to end-answer correctness (as noted by the authors), undermining comparability between vote-based and judge-based methods. Clarify scoring functions and incorporate process-aware verifiers for reasoning-heavy tasks.

- **Underperformance of automated LLM-judge in coding:**
  On IOI 2025, Best-of-*N* (30.79) outperforms 3D (LLM-judge) (18.65), contradicting the claim that the LLM-judge is a reliable general-purpose aggregator. The paper should report judge accuracy against ground truth or human agreement and include prompt variants and verifier-backed judges.

- **Methodological flaws in embodied RL evaluation:**
  The claim of “same token budget” between Best-of-*N* and 3D Human is irrelevant for reinforcement learning, where the dominant cost lies in PPO training, rollouts, and rendering. Compute comparisons should be matched in environment steps or wall-clock time.

- **Statistically unsound reporting of final task scores:**
  Appendix B defines the final task score (FTS) as the maximum across five runs (“we run 5 experiments and report the highest TS as the final task score”), which biases results toward 3D. Means ± confidence intervals (or medians) should be reported instead, with corrections for multiple comparisons.

- **Limited human-preference study:**
  The HumanoidJump human-preference experiment (17/20) involves too few trials and lacks details on randomization, inter-rater reliability, or control conditions. While interesting, the results are not robust.

- **Missing fixed-budget ablations across (C, B, T):**
  To substantiate the “3D scaling” claim, experiments should fix total token budget (including judge and history tokens) while varying decomposition across axes (e.g., high-*C*/low-*B*/low-*T* vs. low-*C*/high-*B*/low-*T*) to demonstrate synergy beyond raw sampling. Such ablations are missing.

**Questions:**

**Compute accounting:**
How exactly is “total thinking budget” computed in Figs. 2–3? Does it include judge tokens (LLM-judge), reflections, and prompt growth across turns? Please provide a token-by-token decomposition for each point on the curves.

**Fixed-budget ablations:**
Can you provide results where total token budget is fixed (including judge + history) and you sweep $(C,B,T)$ decompositions to show that 3D outperforms the best 1D allocation?

**Judge reliability:**
For each domain, what is the LLM-judge agreement with the human judge and with ground-truth correctness? Please provide confusion matrices and ablate judge prompts (Appendix D.2) to demonstrate robustness.

**Coding domain contradiction:**
Why does 3D (LLM-judge) underperform Best-of-$N$ on IOI? Is the judge selecting worse programs than the Best-of-$N$ oracle more often? Please include per-problem judge selection accuracy and error types.

**Physics evaluation:**
You state physics evaluation relies “almost entirely” on the final answer. How then is process correctness enforced in CPHO (unlike IMO), and is the Batch (Vote) baseline meaningful in that setting? Please clarify the evaluation protocol for multi-part derivations.

**Embodied RL methodology:**
- **(a) Compute matching:** Can you provide environment-step–matched or wall-time–matched comparisons for Best-of-$N$ vs. 3D? The token budget claim is not appropriate for RL.
- **(b) Statistics:** Please replace FTS = maximum with mean ± CI (or median/IQR) across seeds for each method. If you keep FTS, justify statistically why taking maxima is appropriate.
- **(c) Human study:** Provide randomization protocol, inter-rater reliability, and effect size for the 17/20 result.

**Negative examples (Sec. 3.1):**
You mention sampling non-selected responses as negative examples. Are these used for any learning in this paper (e.g., preference model, DPO-style)? Or are they unused? Please clarify.

**Generalization of the “3D law”:**
Beyond the three small competition datasets, can you provide larger-scale or public benchmarks (e.g., MATH500, AIME24/25, HumanEval+, MBPP-long) to demonstrate the 3D effect is not dataset-specific.

---

### Official Review · Reviewer_1DJ4 · 2025-10-31

**Soundness:** 2
**Presentation:** 3
**Contribution:** 1
**Rating:** 2
**Confidence:** 3

**Summary:**

The paper presents a framework for test-time scaling in reasoning reinforcement learning (RL) models. It extends the traditional context-length scaling to three dimensions: context, batch, and turn, which it refers to as 3D test-time scaling, integrating these three factors to improve reasoning accuracy. The paper empirically demonstrates the effectiveness of this approach on challenging testbeds, including problems from the IMO, CPHO, and IOI. Furthermore, it explores human-in-the-loop enhancements, showing how human feedback can further improve the performance of these models in reasoning tasks.

**Strengths:**

1.Human-in-the-loop Integration: The incorporation of human feedback adds valuable insights into improving model performance beyond traditional scaling approaches.

2.Clear Structure and Readability: The paper is well-organized, and the results are presented in a clear and digestible manner, making it easy to follow the progression of the experiments and conclusions.

**Weaknesses:**

1.The paper provides a relatively basic analysis of scaling across different dimensions (multi-turn, batch size), with the conclusion stating: “All three scaling methods achieve substantial improvements at small scales but saturate as the scale becomes larger.” This is a fairly conventional observation and aligns with general expectations, which limits the paper's ability to offer new, groundbreaking insights.

2.the paper does not explicitly address how to allocate resources across these three dimensions (context, batch, turn) given a fixed token budget. While the scaling effects for each dimension are discussed, the paper lacks a detailed exploration of how to optimize and distribute the token budget when computational resources are constrained. A more comprehensive approach to token allocation within the context of these scaling methods would provide further depth to the paper.

3.Performance Saturation: The observation that performance improves up to a certain point but then plateaus or degrades with further scaling should be explored more thoroughly. The paper could benefit from insights into how to avoid this saturation effect in practice, possibly through more sophisticated strategies or optimization techniques.

**Questions:**

1.What potential limitations exist when applying human-in-the-loop feedback in test-time scaling?

2.Are there alternative strategies to batch scaling that could be explored to overcome the issues observed with large batch sizes? Additionally, could it be possible to identify the point at which saturation occurs, allowing us to determine how to allocate the budget effectively before performing actual inference?

---

### Meta-Review · Area_Chair_VbYV · 2026-01-08

**Summary:**

This work proposes to consider batch scaling and turn scaling for efficient test-time scaling for LLM, in addition to the conventional context-length scaling. Several concerns were raised by the reviewers; for example, 3 of 4 reviewers (1DJ4, SrUw, WdNs) point out the limited technical contribution, as the proposed idea is more like a simple combination of existing approaches without fundamental breakthrough in model design or algorithm side. Also, as pointed by 1DJ4, how to allocate resources across these three dimensions is a natural question to resolve, the paper lacks a detailed exploration for this. Furthermore, while the human evaluation was the key to the experimental results, lots of details about human evaluation design were missing (SrUw).

**Reviewer Concerns:**

As the authors did not provide any rebuttal, no concern were resolved.

**Reviewer Scores:**

As there is no rebuttal by authors, the reviewers would maintain or decrease their initial scores (2,6,4,2).

---

### Decision · Program_Chairs · 2026-01-26

Reject